# Within-host genetic diversity of extended-spectrum beta-lactamase-producing Enterobacterales in long-term colonized patients

Lisandra Aguilar-Bultet[1,2], Ana B. García-Martín [1,2], Isabelle Vock[1,2], Laura Maurer Pekerman[1,2], Rahel Stadler[1,2], Ruth Schindler[1,2], Manuel Battegay[1,2], Tanja Stadler [3,4], Elena Gómez-Sanz[1,2] & Sarah Tschudin-Sutter [1,2] ✉

Despite recognition of the immediate impact of infections caused by extended-spectrum beta-lactamase (ESBL)-producing Enterobacterales (ESBL-PE) on human health, essential aspects of their molecular epidemiology remain under-investigated. This includes knowledge on the potential of a particular strain to persist in a host, mutational events during colonization, and the genetic diversity in individual patients over time. To investigate long-term genetic diversity of colonizing and infecting ESBL-*Klebsiella pneumoniae* species complex and ESBL-*Escherichia coli* in individual patients over time, we performed a ten-year longitudinal retrospective study and extracted clinical and microbiological data from electronic health records. In this investigation, 76 ESBL-*K. pneumoniae* species complex and 284 ESBL-*E. coli* isolates were recovered from 19 and 61 patients. Strain persistence was detected in all patients colonized with ESBL-*K. pneumoniae* species complex, and 83.6% of patients colonized with ESBL-*E. coli*. We frequently observed isolates of the same strain recovered from different body sites associated with either colonization or infection. Antimicrobial resistance genes, plasmid replicons, and whole ESBL-plasmids were shared between isolates regardless of chromosomal relatedness. Our study suggests that patients colonized with ESBL-producers may act as durable reservoirs for ongoing transmission of ESBLs, and that they are at prolonged risk of recurrent infection with colonizing strains.

Extended-spectrum beta-lactamase (ESBL)-producing Enterobacterales (ESBL-PE) are considered a critical threat by public health authorities, such as the Centers for Disease Prevention and Control (CDC) and the World Health Organization (WHO)[1,2]. Infections with ESBL-PE are associated with excess morbidity, mortality, and higher health-care costs. Despite this recognition of the immediate impact on human health, important aspects regarding the molecular epidemiology of ESBL-PE remain largely under investigated. In particular, there is a lack of knowledge regarding the potential of a particular strain to persist in a host over long time periods, the number of mutational events occurring within a host during colonization, and the genetic diversity of colonizing and infecting isolates within an individual patient and

[1]Division of Infectious Diseases and Hospital Epidemiology, University Hospital Basel, University of Basel, Basel, Switzerland. [2]Department of Clinical Research, University Hospital Basel, University of Basel, Basel, Switzerland. [3]Swiss Institute of Bioinformatics, Lausanne, Switzerland. [4]Department of Biosystems Science and Engineering, ETH Zurich, Basel, Switzerland. ✉e-mail: sarah.tschudin@usb.ch

over time. Such knowledge is critical for a deeper understanding of the ESBL-PE epidemiology requiring data on reservoirs and transmission chains, their sustainability and thus their potential to persist over time. Further, investigation of bacterial in-host mutation events may provide important knowledge on the host-pathogen interactions required to establish successful colonization and subsequently infection. So far, ESBL-PE colonization has been mainly studied in presumably "healthy" returning travelers or patients with limited follow-up periods usually not longer than 24 months[3–6]. In addition, these studies mostly lack detailed whole-genome analyses[3,4,7].

At our institution, patients colonized with ESBL-PE are routinely screened at different body sites upon each admission and at least one ESBL-PE isolate recovered by routine clinical sampling is systematically collected. In this work, we investigate the within-host genetic diversity of colonizing and infecting ESBL-*Klebsiella pneumoniae* species complex and ESBL-*Escherichia coli* isolates and their plasmids by sequencing these isolates, collected over a 10-year period from different screening and clinical samples.

## Results

### Patients

From 2008–2018, 495 known ESBL-carriers were prospectively screened by performance of rectal swabs at each visit. Of these, 73 consecutive patients with isolation of at least two ESBL-PE belonging to the same species (*K. pneumoniae* or *E. coli*) from two consecutive rectal swabs were included in this study. Twelve of them were colonized with ESBL-*K. pneumoniae* species complex, 54 with ESBL-*E. coli*, and seven patients were colonized with both ESBL-PE. The Supplementary Table S1 summarizes the baseline characteristics of all the patients included in this study, stratified by species.

### Bacterial isolates

During the 10-year study period, 360 ESBL-PE isolates were recovered (76 *K. pneumoniae* and 284 *E. coli*) (Supplementary Fig. S1). Among isolates initially classified as *K. pneumoniae* by conventional methods, 70 isolates were confirmed as belonging to *K. pneumoniae sensu stricto*, five were identified as *K. quasipneumoniae* (*K. quasipneumoniae* subsp. *quasipneumoniae* n = 3 and *K. quasipneumoniae* subsp. *similipneumoniae* n = 2) and one as *K. variicola* subsp. *variicola* by sequencing. Virulence analyses done for all *K. pneumoniae* species complex isolates revealed no hypervirulent strains. A detailed summary of the Kleborate results is shown in the Supplementary Data S3.

Most isolates were recovered from screening samples (315/360, 87.5%), the rest (45/360, 12.5%) were recovered from samples collected due to suspected infection (4/76, 5.6% of all *K. pneumoniae* species complex and 41/284, 14.4% of all *E. coli* isolates). The majority of isolates was recovered from rectal swabs (237/360, 65.8%), while the remaining isolates (123/360, 34.2%) were obtained from other body sites.

A median of four isolates per patient (IQR 3–5) were collected (Table 1). The maximum number of isolates collected per patient was eight and 19 for *K. pneumoniae* species complex and *E. coli* (Table 1, Supplementary Fig. S2).

### Bacterial diversity and multi-locus sequence typing results

The overall sequence type (ST) diversity per species was high, with a Simpson's diversity index of 0.929 for *K. pneumoniae* species complex and 0.803 for *E. coli*. Isolates were allocated to 20 different STs of *K. pneumoniae* species complex and 38 different STs of *E. coli*. Among the 20 different STs of *K. pneumoniae* species complex, ST48 (n = 12), ST45 (n = 10), ST307 (n = 8), ST985 (n = 7) and ST394 (n = 5) were the five most abundant ones. In *E. coli*, ST131 (n = 120), ST648 (n = 23), ST10 (n = 18), ST405 (n = 15) and ST362 (n = 14) were the most abundant ones. The full list of STs identified is provided in the Supplementary Data S1. According to the cgMLST analyses, 93.4% of *K. pneumoniae*

**Table 1 | Extended-spectrum beta-lactamase producing Enterobacterales (ESBL-PE) isolates recovered per patient, stratified by species**

| | *Klebsiella pneumoniae* species complex (n = 76) | *Escherichia coli* (n = 284) |
|---|---|---|
| | Median (range) | Median (range) |
| Isolates per patient | 4 (2–8) | 4 (2–19) |
| Time interval between first and last isolate (days) | 445 (73–2211) | 456 (36–3387) |
| Number of STs[a] per patient | 1 (1–4) | 1 (1–5) |
| Number of different strains per patient[b] | 1 (1–4) | 2 (1–7) |
| Number of CPP[c] | 1 (1–2) | 1 (0–2) |

[a]Sequence types (STs).
[b]Different strains were defined as isolates not belonging to the same cgMLST cluster (>15 allelic differences for *K. pneumoniae* species complex and >10 allelic differences for *E. coli*).
[c]Cluster(s) per patient (CPP). Singletons are excluded in these counts. A CPP value of 0 indicates that all isolates are genetically different in this patient, and do not cluster together according to the cgMLST scheme used.

species complex isolates and 80.2% of *E. coli* isolates were grouped into 17 and 53 clusters, respectively.

### cgMLST clusters and strain persistence within the same patient

In 84.9% of patients (62/73) only one cluster of ESBL-PE was identified (Fig. 1). In one patient colonized with *K. pneumoniae* species complex and five patients colonized with *E. coli*, two different clusters were detected (Fig. 1).

Fifteen and 28 patients were colonized with only one strain within ESBL-*K. pneumoniae* species complex and ESBL-*E. coli* isolates, respectively (43/73, 58.9%). The rest of the patients (4 colonized with *K. pneumoniae* species complex and 33 patients colonized with *E. coli*) harbored more than one bacterial strain of each species (Fig. 1). One patient (Pat02) carried six different strains belonging to different species: *E. coli* (two strains from ST648), *K. pneumoniae* (two strains from ST20 and ST422), *K. variicola* (one strain), and *K. quasipneumoniae* (one strain) during an eight-year time frame. Detection of >1 ESBL-PE species (*K. quasipneumoniae* and *E. coli*) at the same hospital admission occurred once.

Persistent carriage of the same strain was detected for 100% of patients colonized with *K. pneumoniae* species complex and 83.6% of patients colonized with *E. coli*. For patients colonized with *K. pneumoniae* species complex the median time between collection of the first and the last isolate was 445 days (IQR 210.5–909); for patients colonized with *E. coli*, 456 days (IQR 258–1230 days) (Table 1, Fig. 2A). Colonization with the same strain persisted for a median of 231 days (IQR 106–535) and 332 days (IQR 170–780) for *K. pneumoniae* species complex and *E. coli*, respectively. Persistent colonization by the same ESBL-*E. coli* and/or *K. pneumoniae* strain lasted for at least one, three and five years in 20.5%, 6.8% and 5.5% of patients, respectively. The longest observed strain persistence was 1704 days for *K. pneumoniae* (Fig. 2B) and 3387 days for *E. coli* (Supplementary Table S2, Fig. 2B).

Comparisons between patients with short-term and long-term persistence of colonization are presented in the Supplementary Data S4 and revealed immunosuppression as being associated with long-term rather than short-term colonization. In addition, comparisons between short- and long-colonizer strains did not reveal any causal relationship with ST classification, nor the presence of any specific ESBL gene or incompatibility group (Supplementary Data S5).

### Diversity among ESBL-PE isolates of the same cgMLST cluster

The median value of maximum allelic differences within the same cgMLST cluster was 3·5 (IQR 1.5–5), for *K. pneumoniae* species complex

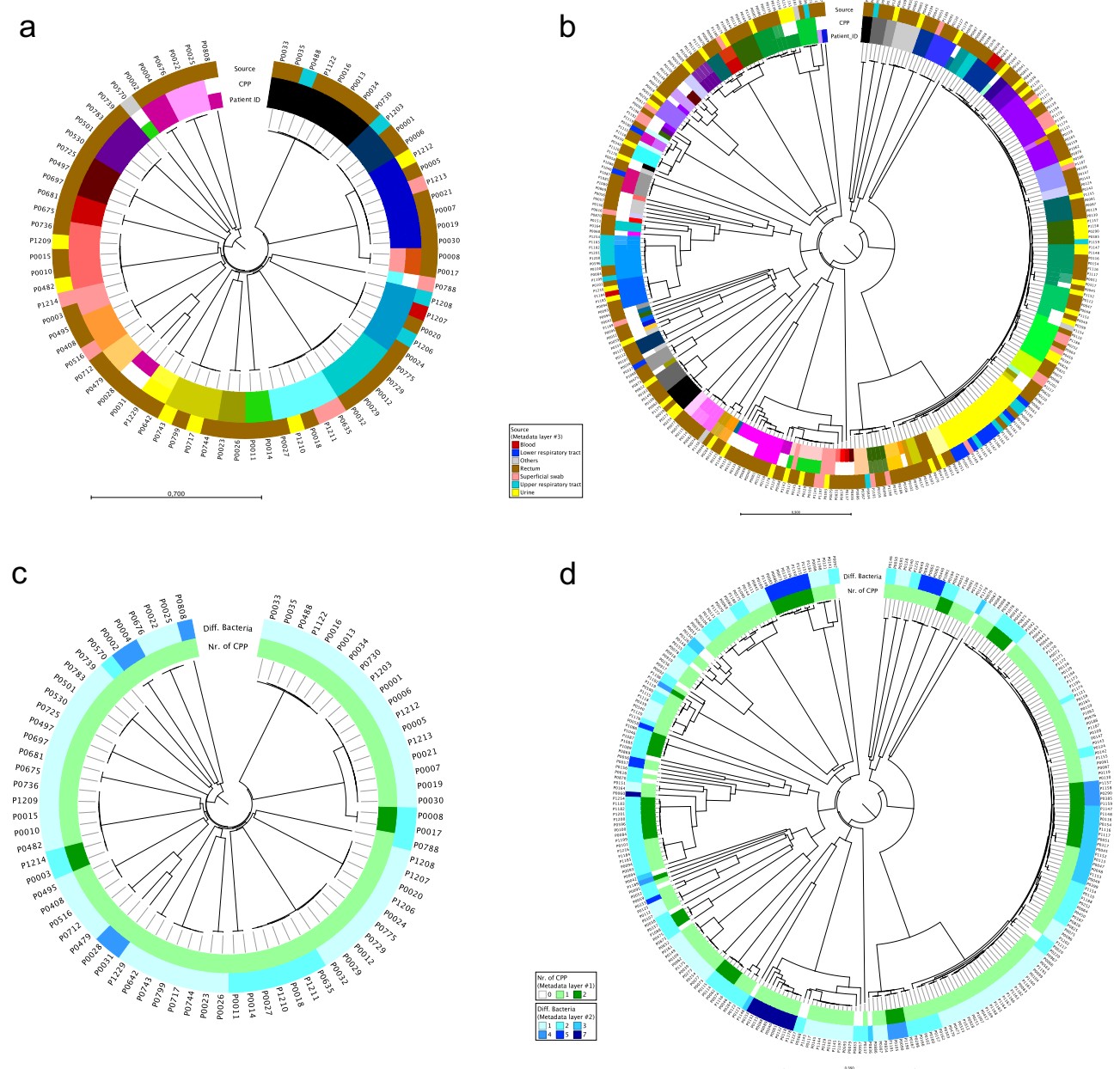

**Fig. 1 | Genetic relatedness of extended-spectrum beta-lactamase-producing Enterobacterales (ESBL-PE) based on cgMLST.** Genetic relatedness of ESBL-PE isolates based on cgMLST. Dendrogram tips correspond to the isolates (named as "P" followed by four digits). **a** *K. pneumoniae species* complex isolates (*n* = 76). Starting from inside, color rings represent patient identification numbers (Patient ID), clusters per patient (CPP) and origin of the sample. **b** *E. coli* isolates (*n* = 284). Starting from inside, color rings represent patient identification numbers (Patient ID), clusters per patient (CPP) and origin of the sample. **c** *K. pneumoniae* species complex isolates (*n* = 76). Starting from inside, number of CPP identified and number of different bacterial strains identified within the same host. **d** *E. coli* isolates (*n* = 284). Starting from inside, number of CPP identified and number of different bacterial strains identified within the same host. Source data are provided as a Source Data file.

and 4 (IQR 1–8) for *E. coli* (Supplementary Figs. S3 and S4). These values were less than half compared to the cut-off values used to define a cluster in the cgMLST schemes used, indicating a high genetic relatedness among isolates of the same cluster. Supplementary Fig. S5 illustrates the overall distribution of the maximum allelic differences per cluster per patient at population level (the whole dataset) versus the delta time of the first and last isolates within the cluster, classified per species.

The SNP analysis considered the whole genome, therefore, it exhibited a higher resolution than the cgMLST-based allelic difference analysis.

To study the long-term within-host diversity, we stratified the data by body sites, focusing on screening samples collected by rectal swabs (median of 315 days for *K. pneumoniae* species complex, ranging from 73 to 1068 days among 53 colonized patients, and median of 233 days for *E. coli*, ranging from 0 to 2841 days among 132 colonized patients) (Fig. 3 and Supplementary Fig. S6). The median number of SNPs within rectal swab isolates of the same strain was 7 (IQR 4.3–9.0) and 7 (IQR 4.0–12.0) for *K. pneumoniae* species complex and *E. coli*, respectively.

The median mutation rate per site per year was 1.5e−06 (IQR, 7.3e −07–3.5e−06) and 1.4e−06 (IQR, 7.6e−07–3.2e−06) for *K. pneumoniae*

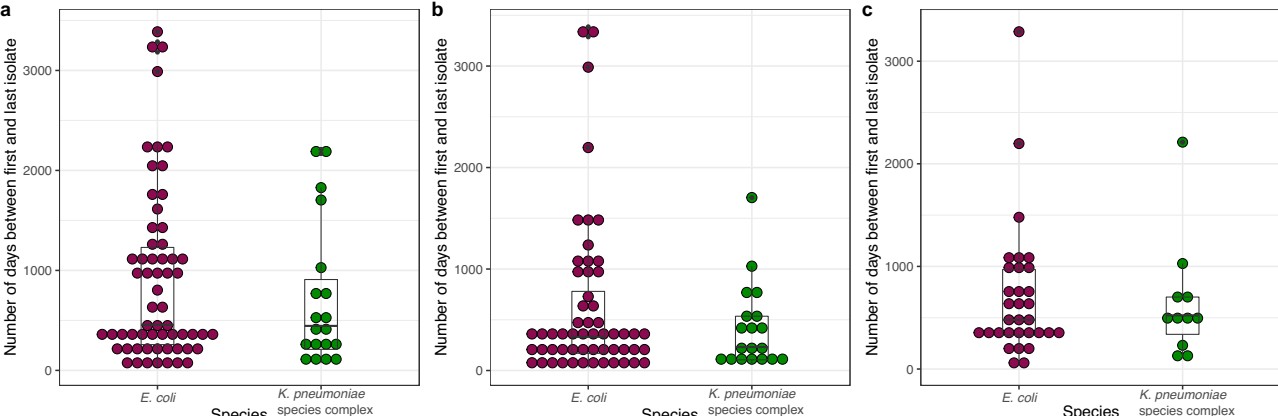

**Fig. 2 | Distribution of the delta time between the first and the last isolate recovered per patient.** Distribution of the delta time between the first and the last isolate recovered per patient. All distinct individual samples (delta time values) are represented by dots. Boxplots grouped the data from first to third quartile, and the median is represented by the black horizontal line inside the bloxplots. Vertical lines outside the boxplots denote the lower and upper extremes. **a** All isolates of a single patient (*n* = 61 delta time-points for *E. coli* and *n* = 19 for K. *pneumoniae* species complex). **b** Isolates exclusively belonging to the same cluster within the same patient (*n* = 56 delta time-points for *E. coli* and *n* = 19 for K. *pneumoniae* species complex). **c** Isolates harboring persistent ESBL-plasmids, despite their chromosomal relatedness (*n* = 32 delta time-points for *E. coli* and *n* = 11 for K. *pneumoniae* species complex). Source data are provided as a Source Data file.

and *E. coli*, respectively. No significant differences were found between the two groups, Mann–Whitney U test with *p*-value = 0.985.

### Diversity and persistence at different body sites

Over the whole study period, three patients (Pat05, Pat09 and Pat13) accommodated not only colonizing but also infecting *K. pneumoniae* species complex isolates (Fig. 4). Two of them (Pat09 and Pat13) were infected by the same colonizing strain (in a time period of five to 18 months) at different body sites (Fig. 4).

Twenty-five patients harbored both colonizing and infecting *E. coli*-ESBL isolates. Twenty-one of them (84%) were infected with the colonizing strain (over a time period ranging from one month until nine years), and four were infected with a different strain than the colonizing one (16%) (Fig. 5). In one patient (Pat48), two different strains were associated with both colonization and infection (in a time period of six months). Six of the 21 patients (Pat22, Pat27, Pat35, Pat43, Pat44 and Pat45) (28.6%) with the same colonizing and infecting strain exhibited recurrent infections. In the four remaining patients with different colonizing and infecting strains, only one infecting isolate was detected in each case (Pat25, Pat30, Pat50 and Pat58).

### Diversity at different body sites during the same time period

Three patients colonized with *K. pneumoniae* and 19 patients colonized with *E. coli* harbored two or more isolates recovered from different body sites within a time window of 90 days.

In the three patients colonized with *K. pneumoniae*, all nine isolates belonged to the same cgMLST cluster within each individual patient with a median allelic difference of 0 (IQR 0–2). In 15 of the 19 patients (78.9%) colonized with *E. coli*, isolates grouped into the same cgMLST cluster with a median allelic difference of 1 (IQR 0–3.5). In six patients, at least one second strain was identified, irrespective of the body site. In four of these patients, different strains were recovered from the same body sites. Details about the different body sites and the exact window time are shown in the Supplementary Data S1.

### Diversity of plasmid types and antimicrobial resistant genes

In both species, the most abundant plasmid incompatibility type was IncF (160 out of 179, 89.4% in *K. pneumoniae* species complex and 565 out of 838, 67.4% in *E. coli*). Within the IncF groups, the majority matched to IncFIB(Kpn3) and IncFII(pKP91) in *K. pneumoniae* species complex, and IncFIB(AP001918), IncFIA and several types of IncFII replicons in *E. coli*. The second most abundant replicon in

*K. pneumoniae* species complex was IncQ and all other replicons found represented a minority (IncI, IncHI and IncL/M). In *E. coli*, IncI was the second most predominant replicon, followed by IncQ and IncX. A detailed list with the replicon types is provided as part of the Supplementary Data S2.

The CTX-M-1 group (specifically *bla*$_{CTX-M-15}$ and *bla*$_{CTX-M-1}$ genes) was the most predominant ESBL group in the two species (Supplementary Fig. S7). An exhaustive list of all the identified AMR genes is part of the Supplementary Data S2.

### Diversity and persistence of acquired resistance genes and Inc replicons

The persistence and transmission of predicted plasmid-associated elements was assessed by comparing the acquired AMR and plasmid replicon profiles (presence/absence of these genes) of all isolates of each patient (Supplementary Figs. S8 and S9). The *oqxA/B* and *mdfA* genes were removed from these plasmid analyses as we observed they were intrinsic to the *K. pneumoniae* and *E. coli* chromosomes, respectively, as described before in other studies[8–12].

Various AMR genes were shared by most isolates of the same patient regardless of belonging or not to the same strain (Table 2). An exhaustive list of all AMR genes identified and the proportion of isolates within the same patient carrying each gene is provided in the Supplementary Data S2.

In *K. pneumoniae* species complex, the most abundant AMR gene was *bla*$_{CTX-M-15}$, which was present in 16 patients (84.2%) and in all isolates recovered from them. In *E. coli*, the most abundant AMR genes were *sul1*, *mph*(A), *aph(6)-Id* and *aadA5*, present in 39 (63.9%), 37 (60.7%), 35 (57.4%) and 35 (57.4%) patients, respectively. The most frequent beta-lactamase genes were *bla*$_{TEM-1b}$ and *bla*$_{CTX-M-15}$, present in 35 (57.4%) and 34 (55.7%) patients. All isolates of these 34 patients harbored *bla*$_{CTX-M-15}$.

In the seven patients colonized with both *K. pneumoniae* species complex and *E. coli* (Pat01, Pat02, Pat05, Pat10, Pat13, Pat17 and Pat18), the ESBL genes shared among isolates of different species in the same patient were *bla*$_{CTX-M-15}$ (shared in four patients involving 26 different isolates), followed by *bla*$_{CTX-M-1}$ and *bla*$_{CTX-M-14}$ (both shared in one patient involving four isolates). Only isolates from different ESBL-PE species from patient Pat18 did not share any ESBL gene (Supplementary Data S2).

In *K. pneumoniae*, IncFIB (Kpn3) and IncFII (pK91) were the most persistent and abundant Inc replicons (Table 3). In *E. coli*, the IncF

a

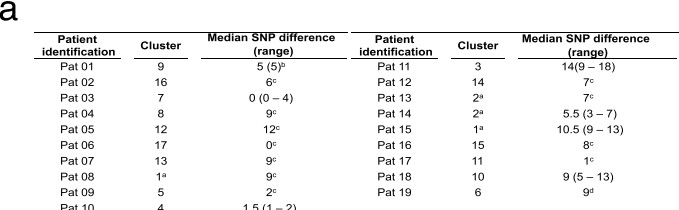

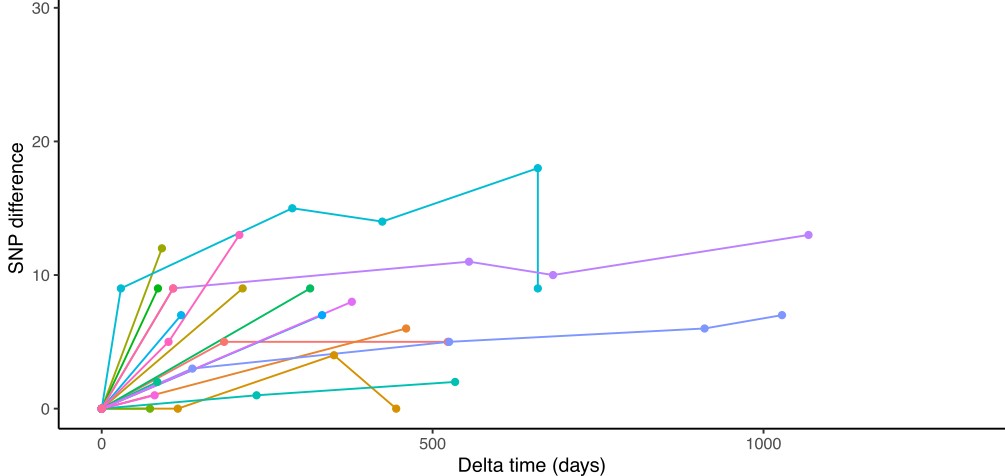

b

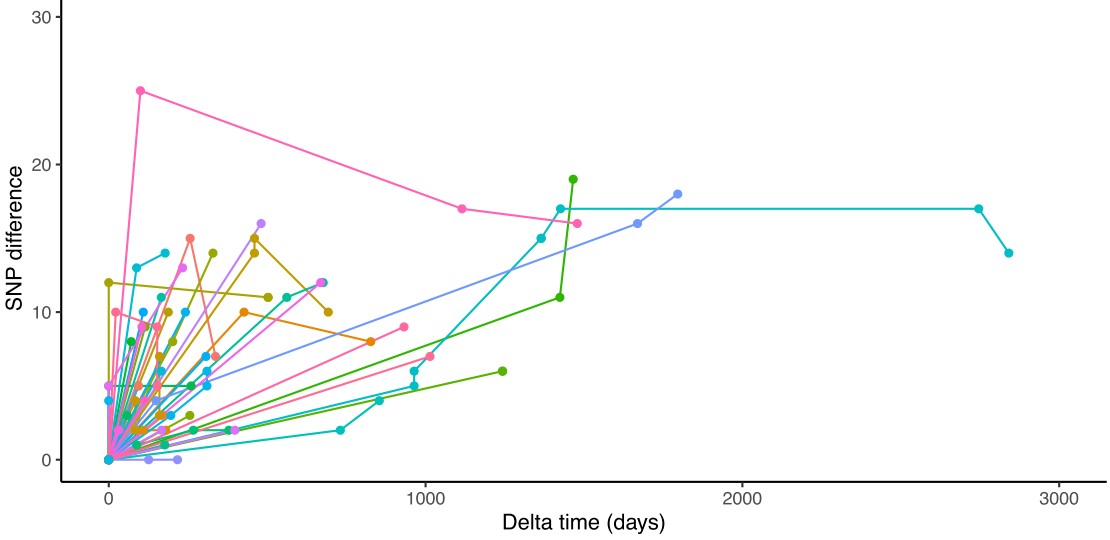

**Fig. 3 | Long-term within-host diversity per patient.** Long-term within-host diversity per patient. Representation of SNPs per cluster and per patient compared against the first isolate of the cluster at each time point. Each line represents a different cluster from a patient. The dots represent the distance in SNPs at each time point with respect to the first (in time) isolate of this cluster, including the comparison of the first isolate against itself represented by the first data point at 0 for each line. An ascending fragment of the line indicates that the following isolate has some SNPs with regard to the first isolate. A descending fragment of the line however indicates that some SNPs of the previous isolate were not kept in the following one. The embedded tables register the median and range of all SNP differences. [a]Same cluster in different patients; [b]Minimum and maximum values are equal; [c]Only two isolates were included in the analysis; thus, only one value of SNP differences is registered. **a** *Klebsiella pneumoniae* species complex. **b** *Escherichia coli*. Source data are provided as a Source Data file.

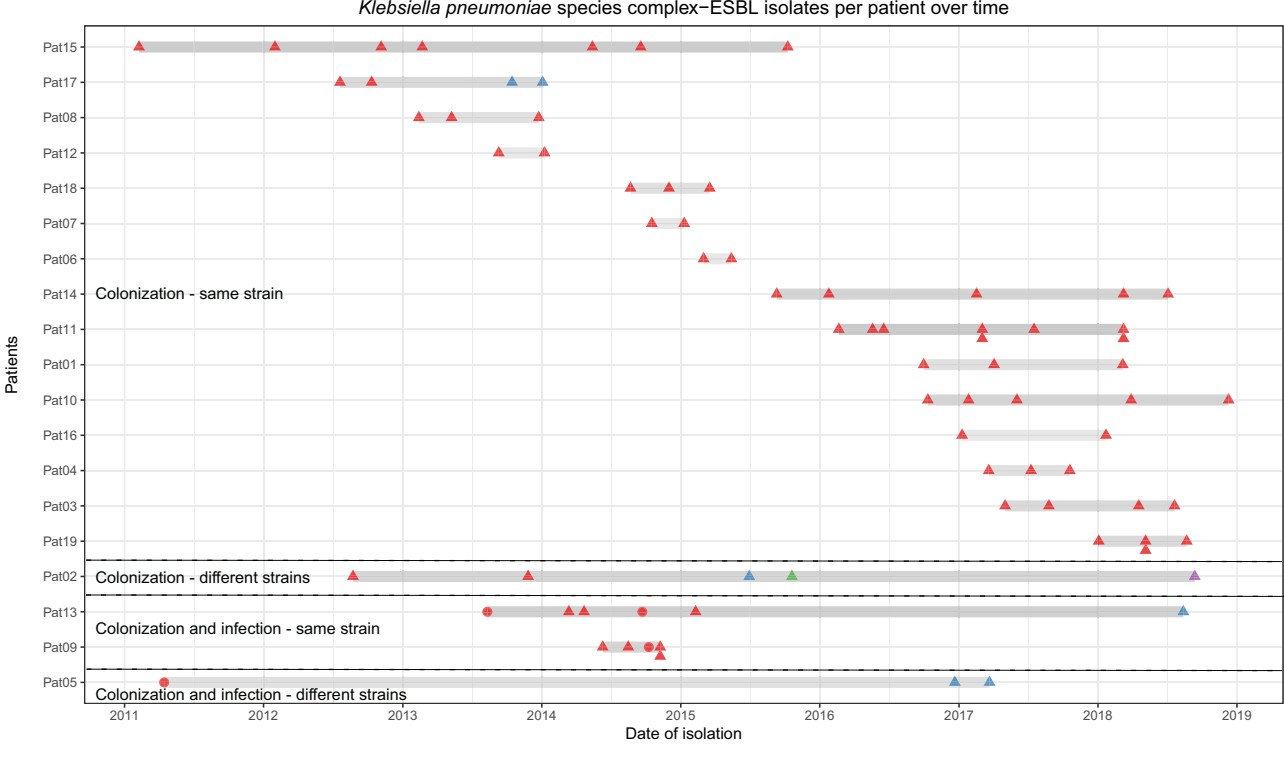

*Klebsiella pneumoniae* species complex−ESBL isolates per patient over time

**Fig. 4 | Carriage of extended-spectrum beta-lactamase-producing *Klebsiella pneumoniae* species complex per patient over time.** ESBL-*Klebsiella pneumoniae* species complex isolates carriage per patient over time (*n* = 76). Gray bars represent the study period per patient including the time between the first and the last isolate. Circles and triangles denote isolates recovered for each patient. Same color per patient indicates isolates belonging to the same cluster (same strain), and different colors indicate different strains. Various isolates collected on the same day are represented as a pile of symbols (circles or triangles) one above the other/s.

groups IncFIB(AP001918) and IncFIA were the most persistent and prevalent replicons, together with IncXA.

Four patients (Pat01, Pat02, Pat05 and Pat13) out of the seven colonized with both *K. pneumoniae* species complex and *E. coli* shared different Inc replicons. The replicon IncI1_1_Alpha was shared among five isolates in two patients, while IncFIB(K)_1_Kpn3 and IncFII_1_pKP91 were shared both in two patients involving 12 isolates (Supplementary Data S2).

In isolates of different species within the same patient, ESBL/Inc pairs such as CTX-M-1/IncI1_Alpha, CTX-M-15/IncFIB(K)_Kpn3, CTX-M-15/IncFII_1_pKP91 and CTX-M-14/ IncI1_1_Alpha were detected, which might suggest horizontal gene transfer of at least part of the ESBL-plasmid.

### Persistence of putative ESBL-plasmids

The genome assemblies obtained from the 30 selected isolates sequenced by PacBio (from 22 different patients) served as references to align all Illumina-based contigs from the rest of isolates of each patient to find out whether all elements of the reference plasmids were kept in the subsequent isolates of the same cluster. Twenty-three ESBL-plasmids were reconstructed (Supplementary Table S3, Source Data file).

In *K. pneumoniae* isolates, eight ESBL-plasmids (median size 194.1 kb, ranging from 83 kb to 267.2 kb) for eight patients were reconstructed (Supplementary Table S3). In all these patients, plasmid persistence was observed in at least two isolates, according to BRIG results (Source Data file). However, in three patients (Pat10, Pat11 and Pat15), deletions of plasmid regions (ranging from 28 kb to 49 kb) were detected in eight isolates, despite belonging to the same strain. In all cases, the ESBL genes and most AMR genes were maintained despite the plasmid differences in other regions.

In *E. coli*, 22 isolates underwent long-read sequencing. Of these, eight cases were excluded from the ESBL-plasmid analysis, since the ESBL gene was located on the chromosome (Supplementary Table S3). For the remaining 14 patients, 15 plasmids were reconstructed (median size 107.8 kb, ranging from 42.8 kb to 139.4 kb) (Source Data file). In one patient (Pat69) two ESBL-plasmids were identified (one IncI1_Alpha and one IncFII), both persisting for 4 years. All ESBL-plasmids in *E. coli* showed persistence in isolates of the same strain. In three patients (Pat27, Pat46 and Pat55), isolates of different strains showed different putative ESBL-plasmids but sharing the same ESBL genes and most Inc replicons. However, isolates recovered from two patients (Pat48 and Pat61) shared the same ESBL-plasmid (99.6 Kb with IncI1_Alpha and $bla_{\text{CTX-M-1}}$ genes) despite belonging to different strains. In patient Pat47, the ESBL-plasmid was shared by three out of the five isolates of the same cgMLST cluster; the other two isolates did not have any known ESBL gene, but the beta-lactamase gene $bla_{\text{CMY-2}}$, flanked by a similar plasmid backbone and the same Inc replicons as the ESBL-plasmid present in the other isolates.

Median persistent carriage of ESBL-plasmids was 499 days (IQR 211–680.3) and 370 days (IQR 326.5–1085.5) for *K. pneumoniae* and *E. coli*, respectively. Both median values are greater than the median persistent carriage for the strains. The longest carriage was almost three years for *K. pneumoniae* and nine years for *E. coli* (Fig. 3C).

### Persistence of whole ESBL-plasmids

Persistence of whole ESBL-plasmids was confirmed at different levels within individual patients: in isolates of the same cgMLST cluster, in isolates of different strains but same species, and in isolates of different species (Supplementary Data S6).

In *K. pneumoniae* species complex patients, except one cluster with one strain harboring the ESBL gene on the chromosome, in all

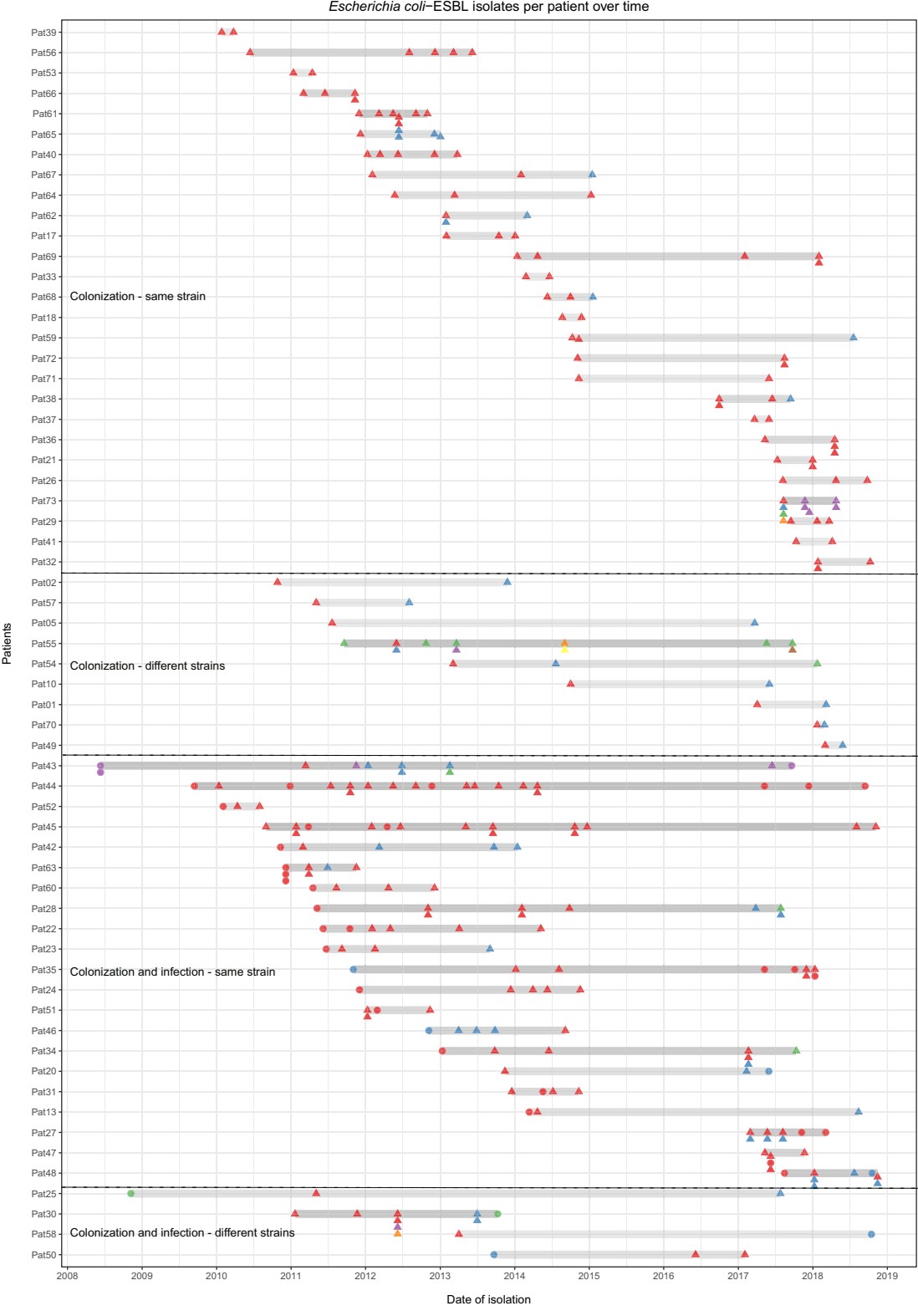

**Fig. 5 | Carriage of extended-spectrum beta-lactamase-producing *Escherichia coli* per patient over time.** ESBL-*Escherichia coli* isolates carriage per patient over time (*n* = 284). Gray bars represent the study period per patient including the time between the first and the last isolate. Circles and triangles denote isolates recovered for each patient. Same color per patient indicates isolates belonging to the same cluster (same strain), and different colors indicate different strains. Various isolates collected on the same day are represented as a pile of symbols (circles or triangles) one above the other/s.

**Table 2 | Antimicrobial resistance genes with median persistence values of 100% (proportion of isolates of each individual patient with detection of a specific gene)**

| Gene | Group | Number of patients harboring an isolate with the indicated gene (%) | Number of isolates with the indicated gene (%) | Species |
|---|---|---|---|---|
| *qnrB1* | Quinolone | 13 (68.4) | 48 (63.2) | *K. pneumoniae* species *complex* |
| *qnrS1* | Quinolone | 4 (21.1) | 11 (14.5) | *K. pneumoniae* species complex |
| *bla*CTX-M-15* | Beta-lactam | 16 (84.2) | 64 (84.2) | *K. pneumoniae* species complex |
| *bla*TEM-1B | Beta-lactam | 15 (78.9) | 58 (76.3) | *K. pneumoniae* species complex |
| *bla*OXA-1 | Beta-lactam | 10 (52.6) | 40 (52.6) | *K. pneumoniae* species complex |
| *bla*SHV-187* | Beta-lactam | 8 (42.1) | 20 (26.3) | *K. pneumoniae* species complex |
| *bla*SHV-106* | Beta-lactam | 2 (10.5) | 12 (15.7) | *K. pneumoniae* species complex |
| *bla*SHV-110 | Beta-lactam | 1 (5.3) | 5 (6.6) | *K. pneumoniae* species complex |
| *bla*OKP-A-12 | Beta-lactam | 1 (5.3) | 3 (3.9) | *K. pneumoniae* species complex |
| *bla*TEM-210 | Beta-lactam | 1 (5.3) | 3 (3.9) | *K. pneumoniae* species complex |
| *aph(6)-Id* | Aminoglycoside | 14 (73.7) | 58 (76.3) | *K. pneumoniae* species complex |
| *aph(3″)-Ib* | Aminoglycoside | 14 (73.7) | 57 (75.0) | *K. pneumoniae* species complex |
| *aac(3)-IIa* | Aminoglycoside | 7 (36.8) | 30 (39.5) | *K. pneumoniae* species complex |
| *ant(3″)-Ia* | Aminoglycoside | 1 (5.3) | 2 (2.6) | *K. pneumoniae* species complex |
| *dfrA14* | Trimethoprim | 13 (68.4) | 55 (72.4) | *K. pneumoniae* species complex |
| *dfrA1* | Trimethoprim | 2 (10.5) | 8 (10.5) | *K. pneumoniae* species complex |
| *dfrA15_2* | Trimethoprim | 1 (5.3) | 2 (2.6) | *K. pneumoniae* species complex |
| *aac(6′)-Ib-cr* | Fluoroquinolone and aminoglycoside | 13 (68.4) | 53 (69.7) | *K. pneumoniae* species complex |
| *tet(A)* | Tetracycline | 14 (73.7) | 52 (68.4) | *K. pneumoniae* species complex |
| *tet(D)* | Tetracycline | 1 (5.3) | 5 (6.6) | *K. pneumoniae* species complex |
| *sul2* | Sulphonamide | 14 (73.7) | 50 (65.8) | *K. pneumoniae* species complex |
| *sul1* | Sulphonamide | 6 (31.6) | 17 (22.4) | *K. pneumoniae* species complex |
| *fosA6* | Fosfomycin | 11 (57.9) | 40 (52.6) | *K. pneumoniae* species complex |
| *fosA_5* | Fosfomycin | 3 (15.8) | 13 (17.1) | *K. pneumoniae* species complex |
| *fosA_3* | Fosfomycin | 3 (15.8) | 6 (7.9) | *K. pneumoniae* species complex |
| *bla*CTX-M-15* | Beta-lactam | 34 (55.7) | 132 (46.5) | *E. coli* |
| *bla*CTX-M-1* | Beta-lactam | 13(21.3) | 50 (17.6) | *E. coli* |
| *bla*CTX-M-14* | Beta-lactam | 9 (14.8) | 40 (14.1) | *E. coli* |
| *bla*CTX-M-14b* | Beta-lactam | 4 (6.6) | 16 (5.6) | *E. coli* |
| *bla*CTX-M-8* | Beta-lactam | 1 (1.6) | 7 (2.5) | *E. coli* |
| *bla*CMY-2 | Beta-lactam | 1 (1.6) | 5 (1.8) | *E. coli* |
| *bla*TEM-190 | Beta-lactam | 1 (1.6) | 5 (1.8) | *E. coli* |
| *aac(3)-IVa* | Aminoglycoside | 1 (1.6) | 8 (2.8) | *E. coli* |
| *aph(4)-Ia* | Aminoglycoside | 1 (1.6) | 8 (2.8) | *E. coli* |
| *catB3_2* | Phenicol | 1 (1.6) | 2 (0.7) | *E. coli* |

Extended-spectrum beta-lactamase (ESBL) genes are marked with an asterisk.

**Table 3 | Plasmid Inc-types with median persistence values of 100% (proportion of isolates of each individual patient with detection of a specific element)**

| Inc-type | Group | Number of patients harboring an isolate with the indicated Inc-type (%) | Number of isolates with the indicated Inc-type (%) | Species |
|---|---|---|---|---|
| IncFIB(K)_Kpn3 | IncFIB | 18 (94.7) | 73 (96.0) | *K. pneumoniae* species complex |
| IncFII_pKP91 | IncFII | 15 (78.9) | 51 (67.1) | *K. pneumoniae* species complex |
| IncFIB(AP001918) | IncFIB | 44 (72.1) | 181 (63.7) | *E. coli* |
| IncFIB(pB171) | IncFIB | 1 (1.6) | 2 (0.7) | *E. coli* |
| IncFIA | IncFIA | 40 (65.6) | 167 (58.8) | *E. coli* |
| IncFII | IncFII | 26 (42.6) | 98 (34.5) | *E. coli* |
| IncX3 | IncX | 5 (8.2) | 23 (8.1) | *E. coli* |
| IncX4 | IncX | 3 (4.9) | 11 (3.9) | *E. coli* |

other clusters ($n = 10$) the isolates selected showed ESBL-plasmid persistence. In *E. coli* patients, similar results were obtained. All ESBL-plasmids were conserved among isolates of the same cgMLST clusters ($n = 23$), except by eight patients with a chromosomal-encoded ESBL gene (as mentioned above), and one strain with a 54 kb ESBL-plasmid (IncN type harboring $bla_{CTX-M-15}$) which was not persistent (Source Data file, Supplementary Data S6).

In *K. pneumoniae* species complex patients, persistence of the same ESBL-plasmid in different strains within the same patient was less likely, only one patient (out of the three with different strains sharing the same ESBL genes and Inc types) showed whole ESBL-plasmid persistence. On the other hand, in 70% of all eligible *E. coli* patients (with different strains harboring the ESBL gene on plasmids), ESBL-plasmid persistence was evidenced. Six patients were excluded since the ESBL gene was detected on the chromosome (Source Data file, Supplementary Data S6).

Out of the seven patients colonized with more than one ESBL-PE species, only two of them followed the criteria to be selected and studied for ESBL-plasmid persistence. In both cases, the persistence of the whole plasmid was confirmed (Source Data file, Supplementary Data S6).

## Discussion

This 10-year longitudinal study provides evidence on the potential for long-term colonization with ESBL-PE in a cohort of patients admitted to an academic tertiary care center, the genetic changes of these microorganisms over time, and the genetic diversity between colonizing and infecting isolates and between different sites of colonization and infection. We estimated strain and plasmid persistence within the same host, and evaluated how isolates of the same cluster/strain changed over time. Further, we estimated plasmid transmission between different strains or species in the same patient.

The same ESBL-PE strain persisted for up to five (*K. pneumoniae*) or nine (*E. coli*) years, which is longer than what has been reported previously for both species[13]. It also differs from reports of ESBL-PE persistence in healthy travelers, in which ESBL-PE carriage was only detected in few patients (11%) after 12 months[3]. While our study design does not allow to draw conclusions regarding the exact duration of colonization, it points to the potential for long-term colonization lasting for months to years in patients with pre-existing medical conditions.

In patients with isolates recovered from different body sites, the same strain was frequently found at different sites and related to both infection and colonization, similar to previous studies[14]. Remaining colonization with the same strain after infection, as well as recurrent infections in patients initially colonized were observed.

Some patients harbored additional ESBL-PE isolates of different genetic backgrounds, as described in other studies[15]. The finding that patients may be colonized with genetically distinct strains of the same species or different species, implies the need to select different colonies for both phenotypic and genotypic testing to inform individual treatment decisions and transmission studies.

In some cases, a higher number of SNPs between the first and the second isolates of the same cluster was observed, unlike in later isolates. This could indicate an earlier colonization of the patient (before the study timeframe), since in cases of recent colonization all isolates would be expected to be more genetically similar. However, during the course of colonization, the number of SNP differences with regard to the first colonizing isolate of the patient remains almost stable. The estimated mutation rates in the order of 10e-06 – 10e-07 are in line with those reported for these species[16–19]. The number of SNPs found in isolates of the same strain within the same patient serves as a basis to establish realistic cut-offs when evaluating patient -to-patient transmission, not only for ESBL-PE[15,20].

In general, plasmid-associated elements were shared between isolates of the same patient even though they were not part of the same strain/cluster, suggesting horizontal gene transfer. The presence of the same set of AMR genes and replicons in all isolates constitutes an additional sign of persistence of AMR-plasmid carriage, including ESBL-plasmids, which was confirmed by long-read sequencing in a subset of isolates. In addition, the reconstruction of ESBL-plasmids from long-read sequencing data allowed to identify transmission of whole ESBL-plasmids between different strains in the same patient.

Our study has several limitations including its single-center design and its non-systematic sampling. Due to the longitudinal design lasting for months to years for individual patients and non-standardized follow-up, information on exposures to antimicrobials could not be assessed. As recovery of ESBL-PE in at least two consecutive rectal swabs was defined as an inclusion criterium of our study and rectal swabs are systematically performed at hospital admission, the study population may be biased towards a "sicker" patient population requiring re-admission. The majority of isolates were collected from rectal swabs, limiting the generalizability to different body sites. As screening of other body sites is part of our institutional policy, the limited number of isolates recovered from other sites, such as the genitourinary tract, reflects persistence mainly occurring in the gastrointestinal tract. As not all colonies were picked for isolation (but only those differing by morphology) the diversity of isolates colonizing or infecting an individual patient may have been under-estimated. Our study design and sample size were only partially suitable to reveal differences in both patient and strain characteristics potentially associated with short-versus long-term colonization. Patients with potential short-term colonization were excluded as they did not fulfill the study's inclusion criteria requiring the detection of colonizing strains from at least two consecutive screening samples. While further studies are needed to address this question, we would like to point to the specific challenge of classifying patients as being colonized over a "short-term" posed by the potential of sampling and detection bias (i.e. representability of screening samples to rule out persistent colonization and the sensitivity of diagnostic approaches to detect persistent colonization). Thus, patients may be classified as colonized over a short time-period based on negative sampling results, despite still being colonized. This is supported by the results of different decolonization studies, revealing short term success, yet long-term failure of such regimens[21]. Our study was underpowered to analyze associations between genome characteristics of colonizing strains and clinical presentation of colonization versus infection. Yet our finding that the majority of strains identified within the context of infection were equal to those identified within the context of colonization, suggests that host rather than bacterial attributes are decisive for transition from colonization to infection. Our study was not designed to investigate potential loss of ESBL-containing plasmids and potential persistence of bacterial strains after loss of ESBL-containing plasmids within individual patients.

The main strength of this study lies in its longitudinal design of a period of ten years and the high number of samples analyzed for some patients allowing to monitor genetic diversity of strains over a long time and at different body sites. Former longitudinal studies investigating detailed genetic changes of colonizing bacteria, and specifically ESBL-PE, ranged mostly from a few months up to one[5,22] or two years[23]. Our design allowed us to study the variability among persistent ESBL-*K. pneumoniae* species complex and ESBL-*E. coli* strains within the same patient and over time, in some cases for up to nine years. Moreover, we generated closed genomes and plasmids for some isolates by long-read sequencing, which served as references for subsequent analyses, and will contribute to the catalog of publicly available genomes. The findings of this study may serve as a valuable basis for further studies designed to analyze strain and host factors related to duration of

colonization and transition from colonization to infection as they provide estimates of the genetic diversity of ESBL-producing *E. coli* and *K. pneumoniae* species complex to be expected within individual patients over time.

In conclusion, the diversity of ESBL-producing *E. coli* and *K. pneumoniae* species complex within patients presenting to a tertiary care center is low and patients may remain colonized with the same strain for up to nine years. Mobile genetic elements are frequently shared among ESBL-PE isolates of the same host. Patients colonized with ESBL-producing *E. coli* and *K. pneumoniae* species complex may serve as durable reservoirs for ongoing transmission of ESBLs and are at risk of recurrent infection with colonizing strains.

## Methods
This study was approved by the local ethics committee (EKNZ-2017 00100) and it is part of the registered NRP project "Transmission of ESBL-producing Enterobacteriaceae" (ClinicalTrials.gov Identifier: NCT03465683)[24]. The local ethics committee did not classify this study as human research but as a quality control project, therefore informed written consent of the study participants was not required. Participants did not receive any compensation.

### Study design and setting
This observational cohort study was performed at the University Hospital Basel, a 735-bed tertiary care center in Basel (Switzerland), which admits approximately 35'000 adult patients annually.

Since 2003, all patients with recovery of ESBL-PE from any specimen obtained by routine clinical practice in both in and outpatient settings are routinely screened to determine further colonization sites. Screening for ESBL-PE carriage is performed by selective plating of rectal swabs, swabs from open wounds or drainages, as well as urine samples from patients with urinary catheters[25].

This study adhered to the Strengthening the Reporting of Observational Studies in Epidemiology (STROBE) guidelines for reporting of observational studies[26].

### Participants and ESBL-PE isolates
Patients admitted to the University Hospital Basel from 01/2008 to 12/2018 with detection of ESBL-PE isolates belonging to the same species (*K. pneumoniae* or *E. coli*) in at least two consecutive rectal swabs were included in this study. Patients with rejection of the general informed consent were excluded.

Colonizing and/or infecting isolates collected from different body sites from patients meeting these inclusion criteria were included. The number of isolates and the time interval between them differ for each patient.

### Data collection
Pertinent clinical and microbiological data were collected retrospectively from electronic medical records and entered into a secured REDCap database. For the collection of follow-up data, information from medical records entered up until June 2023 was considered.

Assessed variables were (1) demographics (2) previous hospitalizations (defined as any hospitalization of at least two days within the past 12 months prior to index hospitalization), (3) travel history (defined as a stay outside of Switzerland within 12 months prior to the index hospitalization), hospitalization abroad (defined as hospitalization outside of Switzerland within 12 months prior to the index hospitalization), (4) comorbidities based on the Charlson Comorbidity Index (CCI), (5) receipt of dialysis during hospitalization, (6) history of organ or (7) allogenic stem cell transplantation, (8) permanent urinary catheterization, (9) proton-pump inhibitor (PPI) and/or other antacid usage concomitant medication, immunosuppressive medication within 12 months prior to index sample, and a known (10) history of ESBL colonization and/or infection within the previous 12 months.

ESBL-PE isolates were classified as colonizing or infecting based on their origin and careful assessment of the patient medical history. Infections with ESBL-PE were defined according to the National Healthcare Safety Network Patient Safety Component Manual by the Centers for Disease Control and Prevention[27].

### ESBL-PE isolation and confirmation
Standard culture methods in accordance with the Clinical and Laboratory Standards Institute (CLSI) guidelines (https://clsi.org/) were applied for detection of ESB-PE in the clinical samples. Chromogenic screening agar plates (chromID ESBL, bioMérieux, Marcy-l'Étoile, France) were used for bacterial culture and isolation of ESBL-PE of the screening samples. For the blood samples, incubation of aerobic and anaerobic blood culture flasks was performed with the Bact/ALERT® 3D and VIRTUO® system (both from bioMérieux, Marcy-l'Étoile, France) either with charcoal or raisins. In all cases species identification was done by MALDI-TOF mass spectrometry (Bruker Daltonics, Bremen, Germany) or Vitek 2™ System (bioMérieux, Marcy-l'Étoile, France). The Vitek 2™ System was also used for antimicrobial susceptibility testing. Testing for ESBL production in all isolates was based on the detection of resistance to cefpodoxime, ceftriaxone, ceftazidime or aztreonam. Phenotypic confirmation of the ESBL test results was conducted by Etest® strips (bioMérieux, Marcy-l'Etoile, France) using cefotaxime, ceftazidime or cefepime, each tested with and without clavulanic acid, or with ROSCO disks (Rosco, Taastrup, Denmark). As our study was performed over a time period of ten years, the specific breakpoints applied changed over time. From 01/2008 to 05/11, minimal inhibitory concentration (MIC) breakpoints were interpreted according to the guidelines of the CLSI, and from 06/11 to 12/2028 breakpoints were interpreted according to the respectively current version of the EUCAST guidelines, ranging from version 1.3 to 8.1 (www.eucast.org). Isolates were confirmed as ESBL-producers when they displayed resistance to at least two of the three tested substances. Indeterminate results were further evaluated using the Eazyplex® SuperBug CRE panel (amplex, Gars Bahnhof, Germany) specific for the detection of ESBL genes of the CTX-M-1 and CTX-M-9 groups. If these genes (selected based on our local epidemiology) were not present, isolates were considered ESBL-negative.

### DNA extraction
Bacterial samples were plated on blood agar plates (Columbia agar with 5% sheep blood, from bioMérieux, Marcy-l'Étoile, France) and incubated at 37 °C overnight. Bacteria were collected with a loop, resuspended in phosphate-buffered saline 1X (37 mM NaCl, 10 mM phosphate, 2.7 mM KCl, pH 7.4) and harvested by centrifugation at $5000 \times g$ for 10 min. Total DNA extraction and purification were carried out by a QIAcube robotic system (QIAGEN) or a similar robot using the QIAamp DNA mini kit (QIAGEN), according to the manufacturer's recommendations. Concentration of the DNA samples was determined by NanoDrop™ One (Thermo Scientific), PicoGreen® (Thermo Scientific), or Qubit 4 (Thermo Scientific).

### Whole-genome sequencing
DNA samples were sequenced using the Illumina® technology (https://www.illumina.com/) on the NextSeq 500/550 platform (150 bp paired-end reads) at Microsynth AG (Balgach, Switzerland), according to the manufacturer's protocols. Nextera XT protocols (Illumina®) were used for genomic library preparation.

To achieve complete genomes consisting of circularized chromosomes and plasmids, 30 isolates were sequenced using the Pacific Biosciences® (PacBio) Sequel I technology (http://www.pacb.com/) at the Lausanne Genomic Technologies Facilities (Lausanne, Switzerland). SMRTbell libraries with barcoded adapters were used for genomic library preparation. The first isolate (according to the chronological date) of the largest clusters per patient (i.e., cgMLST

cluster with at least 4 isolates either from different dates or from different body sites) was selected. The genomes obtained were used as references for further analyses to map Illumina genomes (with BRIG) from the other isolates of the same cluster to assess whether all elements of the original plasmids were kept. To prove maintenance of the whole ESBL-plasmids over time and between different strains and species within the same patient, additional PacBio sequencing of 95 isolates was performed using the Sequel IIe technology with HiFi reads, at the SeqCenter (Pennsylvania, United States of America). Sequencing libraries were prepared following the PacBio SMRTbell® prep kit 3.0 with the SMRTbell® barcoded adapter plate 3.0. Sample selection criteria were the following:

1. To prove the persistence of the whole ESBL-plasmids in the largest clusters (*i.e.*, cgMLST cluster with at least 4 isolates either from different dates or from different body sites), the last isolate of each cluster which harbored all elements of the ESBL-plasmid based on the previous analysis with Illumina data, was sequenced by PacBio.
2. To prove persistence of the same strain in patients with putative "long-persisting" clusters (*i.e.* clusters with delta-time equal or greater than the median delta-time between first and last isolates in all patients), the first and last isolates of these clusters were selected if ESBL genes and Inc groups were maintained.
3. To identify whether the same ESBL-plasmid was present in different strains colonizing the same patient, one isolate per strain (i.e. one isolate per cgMLST cluster, and all individual singletons) that shared the ESBL genes and Inc groups was selected. In isolates belonging to the same cluster, the date of the isolates was considered and those that were more distant in time were selected, to be able to additionally calculate the persistence between different strains in the same patient (in addition to persistence within the same cluster already addressed in the point above).
4. To study whether the same ESBL-plasmid was present in different species of the same patient, one isolate of each species was selected, if they shared ESBL genes and Inc groups

In addition, other shared AMR genes and replicons were checked and taken into account.

### Genome assembly and annotation
Fastp v.0.20.0[28] was applied for quality control, filtering and trimming of Illumina raw sequencing data. De novo assembly was performed with Shovill v.1.0.9 (https://github.com/tseemann/shovill) using SPAdes v.3.13.1 as assembler[29]. Contigs shorter than 500 bp were excluded from the analysis. De novo assembly of PacBio data was performed using Flye v.2.6 or v.2.9[30]. In some specific cases where chromosome/plasmid definition was not clear, a de novo hybrid assembly combining both short (Illumina) and long-sequencing reads (PacBio) was performed with Unicycler v.4.6[31]. Prokka v.1.12 was used for genome annotation[32].

### *Klebsiella* species verification
Since traditional microbiology methods are not reliable for distinguishing the species within this group, isolates belonging to the *K. pneumoniae* species complex (*n* = 76) were analyzed with Kraken2 v.2.0.8[33] and ribosomal Multilocus Sequence Typing (rMLST)[34] for species verification and kleborate v.2.3.2[35] for subspecies assignation. Kleborate was used as well to investigate virulence of the *K. pneumoniae* species complex isolates.

### Assessment of genetic relatedness and diversity
Two approaches were chosen to assess the genetic relatedness of strains: first, multiple-locus sequence typing (MLST) based on seven housekeeping genes was completed in Ridom SeqSphere+ v.6.0 (Ridom, Münster, Germany). This method was used to evaluate the general diversity of the isolates and per patient as it offers a standardized nomenclature for the isolates analyzed[36,37]. Second, core-genome multi-locus sequence typing (cgMLST) was performed with Ridom SeqSphere+ v.6.0 to assess the genetic relatedness among isolates (using the predefined *E. coli* cgMLST and the *K. pneumoniae sensu lato* cgMLST schemes with 2513 and 2358 loci, respectively). Colonization with the same strain (defined in this study as isolates belonging to the same cgMLST cluster) was assumed in patients carrying isolates with ≤10 allelic differences in the case of *E. coli* and ≤15 allelic differences in the case of the *K. pneumoniae* species complex since these are the default cut-off values to define a cluster in the schemes used.

Simpson's index of diversity was calculated to quantify the overall diversity per species using the online tool for quantitative assessment of classification agreement (http://www.comparingpartitions.info/?link=Tool). The sequences of isolates of unknown STs were submitted to BIGSdb-Pasteur platform (https://bigsdb.pasteur.fr/) and EnteroBase (https://enterobase.warwick.ac.uk/).

### Plasmid prediction
Plasmid replicon genes were predicted by ABRicate v.1.0.1 (https://github.com/tseemann/abricate) using the PlasmidFinder database (accessed on July 1st 2021)[38]. Hits with identity and sequence coverage values under 80% were excluded. Based on the in silico prediction results, contigs with plasmid signals were considered as putative plasmid contigs. Plasmid prediction was verified by PacBio long-read sequencing for selected isolates. All ESBL-plasmids (plasmids carrying ESBL genes) obtained by the assembly of long-reads were used in BRIG[39] as references to facilitate the reconstruction of the rest of the plasmids from the putative plasmid contigs by alignment.

### Prediction of acquired antimicrobial resistant genes
Plasmids are enriched in antimicrobial resistant (AMR) genes[40,41], especially AMR genes conferring resistance to clinically important antibiotics among human isolates[42]. Hence, detection of AMR genes may contribute to plasmid content prediction[43]. For this, AMR genes were predicted by ABRicate v.1.0.1 (https://github.com/tseemann/abricate) using the Resfinder database (accessed on July 1st 2021)[43]. Similar to replicon gene prediction, hits with identity and sequence coverage values under 80% were excluded.

### Single nucleotide polymorphism analysis
Diversity and changes over time in isolates belonging to the same cluster were investigated using Snippy v.4.6.0 (https://github.com/tseemann/snippy). For this, the genome of the isolate sampled first according to the chronological order was defined as the reference genome and the reads of all other isolates were mapped to it. The SNPs of the overlapped regions among all isolates of the same cluster were determined using the *snippy-core* command. ClonalFrameML v.1.12[44] was used to identified recombination sites and further SNPs distance values between pair of isolates were calculated with snp-dist v.0.8.2 (https://github.com/tseemann/snp-dists).

### Assessment of strain diversity and persistence
As detailed above, patients were sampled at every hospital visit, and not within a regular timeframe. Hence, different numbers of data points were recovered per patient. The delta time between collection of the first and the last isolate was calculated for each patient. In addition, we calculated the delta time between collection of the first and the last isolate belonging to the same cgMLST cluster per patient to estimate time of colonization with the same strain, defined as persistence of carriage. To analyze the genetic diversity within the same host, the number of clusters present in each patient (CPP, clusters per patient) and the number of strains (different clusters and singletons) were determined based on the results of the cgMLST analysis. Isolates

belonging to different cgMLST clusters were classified as different strains and isolates belonging to the same cgMLST cluster were classified as the same strain. Strain persistence was assumed when isolates of the same cgMLST cluster originated from different sampling time points, reflecting maintenance over time. We defined colonization of different body sites with the same strain as the recovery of isolates belonging to the same cgMLST cluster from samples collected from different anatomic sites. In a fraction of colonizing isolates ($n = 185$ rectal swabs), we evaluated the variability over time by calculating the SNPs with respect to the first isolate in each cluster of the patients. Per-site mutation rate values for both species were estimated by calculating the SNP differences between pairs of isolates and dividing by the delta time between these isolates and the genome size[45].

## Assessment of plasmid diversity and persistence

The distribution and presence/absence pattern of plasmid replicons and AMR genes per species and per patient were calculated based on the ABRIcate predictions on the Illumina draft genomes. To estimate plasmid persistence and diversity within a single host, all isolates of the same patient were analyzed, independent of being classified as belonging to the same cluster and irrespective of the sampling time point. Presence/absence of all plasmid replicons and AMR genes in these isolates recovered from the same patient was assessed to investigate whether they shared the same plasmid-associated genes (plasmid replicons and acquired AMR genes). Intrinsic genes were removed from the analysis when they were exclusively located on the chromosome. As a first approximation, for each replicon or AMR gene, we calculated how many of the isolates recovered from a single patient had the gene, to assess in-host diversity (in isolates from different body sites) and persistence (in isolates collected at different time-points). Secondly, the maintenance pattern of a group of these genes in different isolates of the same patient was visualized as a heatmap.

Finally, in cases with PacBio data was available, after long-read assembly comparison, persistence of the same ESBL-plasmid was confirmed in isolates of the same patient sharing at least 90% of the whole ESBL-plasmid, irrespective of their chromosome relatedness. ESBL-plasmid persistence was assessed at various levels, in line with the selection criteria for PacBio sequencing described in the "Whole-genome sequencing" section.

## Outcomes

The outcomes of this study are: (1) estimates of the diversity of ESBL-producing *E. coli* and/or *K. pneumoniae* within the same host at a given time point and over time, (2) estimates of the potential duration of colonization with the same strain and the same plasmids and ESBL genes, (3) estimates of the potential for plasmid and gene transmission events between different strains/species recovered from the same patient, (4) insights into the number of single-nucleotide polymorphisms (SNPs) accumulated over time in isolates belonging to the same cluster within the same host. The outcomes were assessed by applying the specific approaches outlined in the methods sections assessment of genetic relatedness and diversity, assessment of plasmid diversity and persistence and statistical analyses.

## Statistical analyses

Statistical tests (Mann–Whitney U test, Fisher's exact test) were performed in R (version 4.2.1). We considered two-sided $p$-values < 0.05 as significant. All measurements were taken from distinct biological samples (isolates) or distinct patients.

To compare patient and strain characteristics potentially associated with short-term or long-term persistence of ESBL-PE carriage, the median time between detection of the first and the last isolate of ESBL-PE in any sample collected at our institution was calculated. Patients were assigned to the group of short-term persistence of ESBL-PE carriage if the delta-time was below the median and to the group of long-

term persistence of ESBL-PE carriage if the delta-time was above the median (Supplementary Data S4). Likewise, their respective strains were also classified as short-term colonizers if the delta-time between the first and last isolate of their cgMLST cluster was below the median and long-term colonizers if the delta-time was above the median (Supplementary Data S5). Singletons were categorized as short-term colonizers since they were identified only once in the patients. Patients with missing follow-up screening samples due to not having any further admissions were excluded from these analyses, as duration of colonization could not be estimated. As our data was not normally distributed, we compared differences between the categorical variables of these two groups by performing Fisher's exact test and differences between continuous variables by performing the Mann–Whitney U test.

## Reporting summary

Further information on research design is available in the Nature Portfolio Reporting Summary linked to this article.

## Data availability

All sequencing and sample data from this study can be accessed at the NCBI database under the BioProject number PRJNA910977. Source Data file can be accessed at: https://doi.org/10.5281/zenodo.10116672. Source data are provided with this paper.

## Code availability

Details of the commands used in the bioinformatics analyses are listed in Supplementary Table S5.

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

## Acknowledgements

This study was funded by the Antimicrobial Resistance National Research Programme (NRP72) from the Swiss National Science Foundation, the Swiss National Science Foundation grants 167060 and 197901, the University of Basel and the University Hospital Basel. Computational calculations were performed at sciCORE (http://scicore.unibas.ch/) scientific computing center at University of Basel. We thank the Lausanne Genomic Technologies Facility (https://wp.unil.ch/gtf/) for offering library preparation, PacBio sequencing and preliminary data analyses of a pilot set of seven isolates at no cost. We are grateful to Prof. Adrian Egli for the critical revision of the article and collaboration with the sample collection process, to the technician team of the Division of Clinical Bacteriology and Mycology of the University Hospital Basel for lab assistance, and to Yandy Abreu-Jorge for helping with the creation of some graphs.

## Author contributions

L.A.-B. contributed to the conception of the study, performed sample preparation, data analyses and interpretation of results, created figures and tables, and wrote the manuscript. A.B.G.-M. contributed to generation of figures and tables, interpretation of results and critically revised the manuscript. I.V. contributed to data collection and revised the manuscript. L.M.P. contributed to sample preparation, data collection and revised the manuscript. R.S. contributed to data collection and revised the manuscript. R.S. contributed to data collection and revised

the manuscript. M.B. critically revised the manuscript. T.S. critically revised the manuscript and collaborated with the funding acquisition. E.G.-S. contributed to interpretation of results and critically revised the manuscript. S.T.-S. obtained funding, conceived and supervised the study, contributed to interpretation of results and critically revised the manuscript.

## Competing interests

The authors declare no competing interests.
