## [Peer Review File · Nature Communications]

REVIEWER COMMENTS

Reviewer #1 (Remarks to the Author):

Aguilar-Bultet et al. described the within-host genetic diversity of ESBL-producing *K. pneumoniae* and *E. coli* isolated from patients in a Swiss hospital. The study included 76 ESBL-*K. pneumoniae* complex and 284 ESBL-*E. coli* isolates collected from 19 and 61 patients. Isolates were recovered from both colonization and infection samples, and strain and plasmid persistence were detected.

However, the paper mainly focuses on description, and limited novel information was revealed. Importantly, the study fails to address clinically important questions, such as the genetic factors that underpin the long-term persistence of ESBL-PE, the factors contributing to infections from colonization, and the extent of colonized strains in promoting the transmission of ESBL into clinical infection isolates, etc. The findings of different strains detected from the same patients or horizontal transfer of AMR plasmids are somewhat expected and do not necessarily provide new insights. Therefore, further in-depth analysis is necessary to reveal different factors affecting colonization, infection, persistence, and non-persistence.

Moreover, the study only analyzed the ESBL isolates, while the non-ESBL isolates, which may have lost the ESBL plasmids, were not considered. It is worth noting that the sequencing data is not yet released. Finally, SNPs located within the repeated regions are supposed to be filtered during SNP analysis, as these regions were hard to resolve by short-read sequencing (lines 414-417).

Reviewer #2 (Remarks to the Author):

The study by Aguilar-Bultet et al. investigates the long-term genetic diversity of colonizing and infecting ESBL-producing *K. pneumoniae* species complex/*E. coli* over a ten-year period. The study is an observational cohort study of 73 patients with isolation of ESBL-PE from two consecutive rectal swabs. In total 360 ESBL-PE were recovered. Overall this is a comprehensive and well performed study that provides important information on the long-term colonisation of ESBL-producing *E. coli* and *K. pneumoniae* species complex. A main strength is the length of follow-up compared to other colonisation studies.

Major comment:

* My main comment is the ESBL plasmid analysis which has limitations that the authors should more clearly take into consideration. The authors describe “Persistence of whole ESBL plasmids”. However, BRIG analysis has limitations in that it only provides information that regions with a defined identity to the closed plasmid reference sequence are present in the short-read sequence data for the other isolates. It is not possible to accurately state that it is the same plasmid. Thus, the analysis is persistence of elements of the reference plasmid and not on whole plasmids. To do this all isolates should have been subjected to long-read sequencing.

Minor comments:

* STs for isolates with an unknown ST should be obtained by submitting these to the respective MLST databases. It is a community effort to expand these.

* Hypervirulence in *Klebsiella* is a great concern. I would suggest that the authors investigate if any of the *Klebsiella* isolates are hypervirulent. This can be done using the Kleborate tool (<https://github.com/klebgenomics/Kleborate>).

* Line 70-72: Please specify if the “*K. pneumoniae* species” isolates are all *K. pneumoniae* sensu stricto, and which subspecies the “*K. quasipneumoniae*” isolates belong to (*similipneumoniae* or *quasipneumoniae*). This can be investigated using Kleborate.

* Line 74: The term “*K. pneumoniae* complex” should be “*K. pneumoniae* species complex” to more accurately follow the existing literature. Please change throughout the manuscript and in the Supplementary material.

* Line 83: Check Table numbering – Table 2 does not show STs, but AMR genes. Is the ST Table missing from the manuscript?

* Line 90-91: Is the reference to Fig. 2 correct. I am not able to identify the patients with different clusters in Fig. 2 which basically only show number of days between first and last isolate in each patient.

* Lines 175-176: *oqxA* and *oqxB* is intrinsic genes in *K. pneumoniae* thus the finding of these on the chromosome is not a major finding. Consider to restrict the analysis to acquired resistance genes. It should also be checked if *mdf(A)* is intrinsic genes in *E. coli* and *K. pneumoniae*.

* Lines 187-188: Table 4 = Table 3? Check labelling of Figures and Tables throughout.

* Lines 349-350: Since breakpoints can change the specific version of the EUCAST breakpoint table used should be specified.

* Lines 367-368: Specify the number of isolates sequenced using PacBio.

* Line 399-400: Specify the version of PlasmidFinder database used.

* Line 407-408: Specify the version of ResFinder database used.

* Supplementary material: There is two Tables labelled with “Table S1”

Point by point response to the Reviewer comments (NCOMMS-23-14102).

Reviewer #1 (Remarks to the Author):

Aguilar-Bultet et al. described the within-host genetic diversity of ESBL-producing *K. pneumoniae* and *E. coli* isolated from patients in a Swiss hospital. The study included 76 ESBL-*K. pneumoniae* complex and 284 ESBL-*E. coli* isolates collected from 19 and 61 patients. Isolates were recovered from both colonization and infection samples, and strain and plasmid persistence were detected.

However, the paper mainly focuses on description, and limited novel information was revealed. Importantly, the study fails to address clinically important questions, such as the genetic factors that underpin the long-term persistence of ESBL-PE, the factors contributing to infections from colonization, and the extent of colonized strains in promoting the transmission of ESBL into clinical infection isolates, etc.

Response:

We appreciate the thoughtful comments and suggestions provided by the reviewer. We acknowledge the concerns raised regarding the focus and novelty of our manuscript.

Regarding focus: the aim of our study was to investigate the within-host genetic diversity of colonizing and infecting ESBL-*Klebsiella pneumoniae* species complex (KpSC) and ESBL-*Escherichia coli* isolates and their plasmids. We recognize the importance of the additional questions raised by the reviewer, *i.e.* analyzing genetic factors underpinning long-term persistence of ESBL-PE and contributing to infections arising from colonizing strains, yet consider our study design and sample size only partially suitable to address these. To investigate within-host genetic diversity of colonizing ESBL-PE-strains over time, only patients providing strains recovered from at least two consecutive rectal swabs were included in this study. Screening for detection of persistent carriage is performed on re-admission. Thus, per definition of our inclusion criteria, patients with detection of ESBL-PE during just one hospital-stay without re-admission to our institution or with re-admission and negative follow-up samples, potentially contributing to the comparator group of “short-term”-persistence were not included. We would like to point to the specific challenge of classifying patients as being colonized over a “short-term” posed by the potential of sampling and detection bias (*i.e.* representability of screening samples to rule out persistent colonization and the sensitivity of diagnostic approaches to detect persistent colonization). Thus, patients may be classified as colonized over a short time-period based on negative sampling results, despite still being colonized. This is supported by the results of different decolonization studies, revealing short term success, yet long-term failure of such regimens.¹ Taking these limitations into account, we performed additional data collection and analyses to partially address this reviewer’s concern. By additional medical chart review of all patients included in this study, we identified patients re-admitted to our institution with screening samples testing negative for ESBL-PE and no further detection of ESBL-PE within any follow-up sample. Patients with missing follow-up screening samples due to not having any further admissions were excluded from these analyses, as duration of colonization could not be estimated. To lengthen the follow-up period, the study period was extended from 2008-2018 to June 2023. Based on the median time between detection of the

first and the last isolate of ESBL-PE in any sample collected at our institution, we categorized patients into having short-term persistence of ESBL-PE carriage (delta-time below the median) and long-term persistence of ESBL-PE carriage (delta-time above the median). We compared the two groups (*i.e.* patients with short- and long-term ESBL-PE colonization in terms of their clinical characteristics and the characteristics of their colonizing strains. Among patient characteristics, we identified exposure to immunosuppression as being related to longer duration of colonization. In terms of strain-related genetic factors potentially distinguishing between short and long duration of colonization, we focused on the most frequent sequence types (ST), plasmid incompatibility types and ESBL-genes, as sample sizes for individual genetic features are low as strains could be allocated to wide diversity of STs, plasmid incompatibility types and genes within our study.

We added the methodology applied to the Methods section and included the results to the supplementary material (based on the limitations mentioned, we did not include these findings into the results section of the main manuscript). We point to these results within the results section and added these considerations to the limitation section of our manuscript. The respective sections in the manuscript now read as follows:

Methods section:

“For the collection of follow-up data, information from medical records entered up until June 2023 was considered.” (page 16, lines 352-353).

“Statistical analyses

To compare patient and strain characteristics potentially associated with short-term or long-term persistence of ESBL-PE carriage, the median time between detection of the first and the last isolate of ESBL-PE in any sample collected at our institution was calculated. Patients were assigned to the group of short-term persistence of ESBL-PE carriage if the delta-time was below the median and to the group of long-term persistence of ESBL-PE carriage if the delta-time was above the median. Likewise, their respective strains were also classified as short-term colonizers if the delta-time between the first and last isolate of their cgMLST cluster was below the median and long-term colonizers if the delta-time was above the median. Singletons were categorized as short-term colonizers since they were identified only once in the patients. Patients with missing follow-up screening samples due to not having any further admissions were excluded from these analyses, as duration of colonization could not be estimated. We compared differences between the categorical variables of these two groups by performing Fisher’s exact test, using R (version 4.2.1).” (page 24, lines 529-540).

Results section:

Comparisons between patients with short-term and long-term persistence of colonization are presented in the Supplementary material and revealed immunosuppression as being associated with long-term rather than short-term colonization. (page 6, lines 113-115).

Discussion section:

“Our study design and sample size were only partially suitable to reveal differences in both patient and strain characteristics potentially associated with short-versus long-term colonization. Patients with potential short-term colonization were excluded as they did not fulfill the study’s inclusion criteria requiring the detection of colonizing strains from at least two consecutive screening samples. While further studies are needed to address this question, we would like to point to the specific challenge of classifying patients as being colonized over a “short-term” posed by the potential of sampling and detection bias (i.e. representability of screening samples to rule out persistent colonization and the sensitivity of diagnostic approaches to detect persistent colonization). Thus, patients may be classified as colonized over a short time-period based on negative sampling results, despite still being colonized. This is supported by the results of different decolonization studies, revealing short term success, yet long-term failure of such regimens.”(page 14, lines 298-308).

Regarding the reviewer’s comment on *“factors contributing to infections from colonization, and the extent of colonized strains in promoting the transmission of ESBL into clinical infection isolates”*, we acknowledge that our study is underpowered to perform such analyzes. Among patients fulfilling this study’s inclusion criteria and colonized with ESBL- *K. pneumoniae* species complex, only three accommodated infecting isolates, among patients colonized with ESBL-*E. coli*, 25 harbored infecting isolates. We would, however, like to respectfully highlight that a whole section of our results is dedicated to describing the diversity of colonizing and infecting isolates within the same patient at different body sites. Time intervals between detection of colonization and infection were carefully analyzed and presented in Figure 5. We consider these results of being of value to the scientific community. To address this reviewer’s comment, we added the following to the limitation section of the manuscript: *“Our study was underpowered to analyze associations between genome characteristics of colonizing strains and clinical presentation of colonization versus infection. Yet our finding that the majority of strains identified within the context of infection were equal to those identified within the context of colonization, suggests that host rather than bacterial attributes are decisive for transition from colonization to infection.”* (page 14, lines 308-312).

Regarding novelty: To the best of our knowledge, there is little data on detailed genetic analyses of ESBL-PE strains colonizing and infecting patients over a time-period of up to 10 years, thus we consider our results of being of value to the scientific community. We believe that performing additional long-read sequencing of 95 strains utilizing the PacBio Sequel II platform in response to the comments provided by the reviewers have further enhanced both the quality and novelty of our results.

The findings of different strains detected from the same patients or horizontal transfer of AMR plasmids are somewhat expected and do not necessarily provide new insights.

Therefore, further in-depth analysis is necessary to reveal different factors affecting colonization, infection, persistence, and non-persistence.

Response:

We agree with the reviewer that these findings could be expected; however, to the best of our knowledge, there is little published data so far supporting these expectations. This is why we designed and executed a study to evaluate the within-host diversity not only at strain level but also at plasmid level. While we acknowledge the limitations of our study in terms of revealing different factors affecting colonization, infection, persistence and non-persistence (as detailed in our responses above), we believe that our study provides important and detailed insights, on which further studies designed to address such questions can be based on. The strengths of our study include the number of patients and their respective isolates analyzed over a long time period by in-depth analyses of their genomes at high resolution, now enhanced by including long-read sequences of a larger number of strains. To further address this reviewer's comment, we included the following in the discussion section of our manuscript: *"The findings of this study may serve as a valuable basis for further studies designed to analyze strain and host factors related to duration of colonization and transition from colonization to infection as they provide estimates of the genetic diversity of ESBL-producing E. coli and K. pneumoniae species complex to be expected within individual patients over time."* (page 15, lines 322-326).

Moreover, the study only analyzed the ESBL isolates, while the non-ESBL isolates, which may have lost the ESBL plasmids, were not considered.

Response: We acknowledge that our study was not designed to address the question of loss of ESBL-containing plasmids and potential persistence of bacterial strains after loss of ESBL-containing plasmids within individual patients. To address this additional research question, screening samples would have had to be performed to isolate non-ESBL-containing strains of *E. coli* and *K. pneumoniae species complex*. To address the reviewer's comment, we added the following to the limitation section of our manuscript: *"Our study was not designed to investigate potential loss of ESBL-containing plasmids and potential persistence of bacterial strains after loss of ESBL-containing plasmids within individual patients."* (page 14, lines 312-314).

It is worth noting that the sequencing data is not yet released.

Response: We thank the reviewer for trying to access the sequencing data. This data is still not publicly accessible but will be made public once the manuscript is accepted for publication. For the revision process, we are glad to provide the following access link:

[https://urldefense.com/v3/_https://dataview.ncbi.nlm.nih.gov/object/PRJNA910977?reviewer=mvgd2r4pfk183vsqs_ebc3ee2ug_!!EDSXHx-qgdzzoNk!oIpQJtdFJ78btsHMfWboggGtzPFVebD4J5apoAfLE7QWcwFW-GFjp6GuPjkxeoqKN1dEG9iEMe1g5KhHVAm80E0VQnRi\\$](https://urldefense.com/v3/_https://dataview.ncbi.nlm.nih.gov/object/PRJNA910977?reviewer=mvgd2r4pfk183vsqs_ebc3ee2ug_!!EDSXHx-qgdzzoNk!oIpQJtdFJ78btsHMfWboggGtzPFVebD4J5apoAfLE7QWcwFW-GFjp6GuPjkxeoqKN1dEG9iEMe1g5KhHVAm80E0VQnRi$)

Finally, SNPs located within the repeated regions are supposed to be filtered during SNP analysis, as these regions were hard to resolve by short-read sequencing (lines 414-417).

Response: Thank you very much for this valuable suggestion. We agree that this a useful approach for this type of SNP analyses. However, we initially wanted to explore all possible SNPs (even those located at repetitive regions) accumulated in the subsequent isolates taking the first one as a reference. In addition, since all isolates belong to the same cgMLST cluster, big differences are not expected. Nevertheless, to address this reviewer's suggestion, we repeated the SNP analyses. Taking the first isolate of the cgMLST cluster as a reference, SNPs were called by mapping the short-reads sequencing data with Snippy v.4.6.0 (<https://github.com/tseemann/snippy>), followed by recombination sites detection step with ClonalframeML v.1.12 (<https://github.com/xavierdidelot/clonalframeml/wiki>). We applied this approach to redo the SNP analyses and to further calculate mutation rates. Applying this approach resulted in the reduction of outliers and slightly differing estimates of SNP differences and mutation rates. We revised the respective sections, as follows:

Methods: *"ClonalFrameML v.1.12 was used to identified recombination sites and further SNPs distance values between pair of isolates were calculated with snp-dist v.0.8.2 (<https://github.com/tseemann/snp-dists>)."* (page 21, lines 479-480).

Results: *"To study the long-term within-host diversity, we stratified the data by body sites, focusing on screening samples collected by rectal swabs (median of 315 days for K. pneumoniae species complex, ranging from 73-1068 days among 53 colonized patients, and median of 233 days for E. coli, ranging from 0-2841 days among 132 colonized patients) (Fig. 4 and Supplementary Fig. S6). The median number of SNPs within rectal swab isolates of the same strain was 7 (IQR, 4.3 – 9.0) and 7 (IQR, 4.0 – 12.0) for K. pneumoniae species complex and E. coli, respectively.* (pages 6-7, lines 129-136).

The median mutation rate per site per year was 1.5e-06 (IQR, 7.3e-07 – 3.5e-06) and 1.4e-06 (IQR, 7.6e-07 – 3.2e-06) for K. pneumoniae species complex and E. coli, respectively." (page 7, lines 145-148).

Reviewer #2 (Remarks to the Author):

The study by Aguilar-Bultet et al. investigates the long-term genetic diversity of colonizing and infecting ESBL-producing *K. pneumoniae* species complex/*E. coli* over a ten-year period. The study is an observational cohort study of 73 patients with isolation of ESBL-PE from two consecutive rectal swabs. In total 360 ESBL-PE were recovered. Overall this is a comprehensive and well performed study that provides important information on the long-term colonisation of ESBL-producing *E. coli* and *K. pneumoniae* species complex. A main strength is the length of follow-up compared to other colonisation studies.

Major comment:

***My main comment is the ESBL plasmid analysis which has limitations that the authors should more clearly take into consideration. The authors describe “Persistence of whole ESBL plasmids”. However, BRIG analysis has limitations in that it only provides information that regions with a defined identity to the closed plasmid reference sequence are present in the short-read sequence data for the other isolates. It is not possible to accurately state that it is the same plasmid. Thus, the analysis is persistence of elements of the reference plasmid and not on whole plasmids. To do this all isolates should have been subjected to long-read sequencing.**

Response: We agree with this reviewer’s comment and performed additional PacBio sequencing of 95 bacterial isolates to draw more accurate conclusions regarding potential plasmid persistence. We used the additional long read sequencing data to confirm predictions based on short-read sequencing data regarding persistence of plasmids within individual colonizing strains and potential transmission of plasmids between strains and species. We therefore selected strains additionally submitted for long read sequencing by PacBio based on the following criteria:

1. To address the question of plasmid persistence within a colonizing strain, we, in addition to the first isolate for which long-read sequencing was already performed, selected the last isolate belonging to an individual bacterial cluster based on the cluster size (i.e. containing at least 4 isolates) or colonization being classified as being “long-persisting” (i.e. clusters with delta-time equal or greater than the median delta-time between first and last isolates in all patients)
2. To identify whether the same ESBL-plasmid was present in different strains colonizing the same patient, we selected one isolate per strain (one isolate per cgMLST cluster, and all individual singletons) that shared the ESBL genes and Inc groups and other replicons (according to the AMR identification in Illumina data). In isolates belonging to the same cluster, we also considered the date of the isolates and selected those that are more distant in time, to be able to additionally calculate the persistence between different strains in the same patient (in addition to persistence within the same cluster already addressed in the point above).
3. To study whether the same ESBL-plasmid was present in different species of the same patient, we selected one isolate of each species if they shared ESBL genes and replicons.

The Methods and Results sections were modified accordingly and now read as follows:

Methods section (page 18, lines 399-428): *“To achieve complete genomes consisting of circularized chromosomes and plasmids, 30 isolates were sequenced using the Pacific Biosciences® (PacBio) Sequel # I technology (<http://www.pacb.com/>) at the Lausanne Genomic Technologies Facilities (Lausanne, Switzerland). SMRTbell libraries with barcoded adapters were used for genomic library preparation. The first isolate (according to the chronological date) of the largest clusters per patient (i.e., cgMLST cluster with at least 4 isolates either from different dates or from different body sites) was selected. The genomes obtained were used as references for further analyses to map*

Illumina genomes (with BRIGG) from the other isolates of the same cluster to assess whether all elements of the original plasmids were kept. To prove maintenance of the whole ESBL-plasmids over time and between different strains and species within the same patient, additional PacBio sequencing of 95 isolates was performed using the Sequel IIe technology with HiFi reads, at the SeqCenter (Pennsylvania, United States of America). Sequencing libraries were prepared following the PacBio SMRTbell® prep kit 3.0 with the SMRTbell® barcoded adapter plate 3.0. Sample selection criteria were the following:

1. To prove the persistence of the whole ESBL-plasmids in the largest clusters (i.e., cgMLST cluster with at least 4 isolates either from different dates or from different body sites), the last isolate of each cluster which harbored all elements of the ESBL-plasmid based on the previous analysis with Illumina data, was sequenced by PacBio.

2. To prove persistence of the same strain in patients with putative “long-persisting” clusters (i.e. clusters with delta-time equal or greater than the median delta-time between first and last isolates in all patients), the first and last isolates of these clusters were selected if ESBL genes and Inc groups were maintained.

3. To identify whether the same ESBL-plasmid was present in different strains colonizing the same patient, one isolate per strain (i.e. one isolate per cgMLST cluster, and all individual singletons) that shared the ESBL genes and Inc groups was selected. In isolates belonging to the same cluster, the date of the isolates was considered and those that were more distant in time were selected, to be able to additionally calculate the persistence between different strains in the same patient (in addition to persistence within the same cluster already addressed in the point above).

4. To study whether the same ESBL-plasmid was present in different species of the same patient, one isolate of each species was selected, if they shared ESBL genes and Inc groups

In addition, other shared AMR genes and replicons were checked and taken into account.”

Results section (page 11, lines 239-255):

“Persistence of whole ESBL-plasmids

Persistence of whole ESBL-plasmids was confirmed at different levels within individual patients: in isolates of the same cgMLST cluster, in isolates of different strains but same species, and in isolates of different species.

In *K. pneumoniae* species complex patients, except one cluster with one strain harboring the ESBL gene in the chromosome, in all other clusters (n=10) the isolates selected showed ESBL-plasmid persistence. In *E. coli* patients, similar results were obtained. All ESBL-plasmids were conserved among isolates of the same cgMLST clusters (n=23), except by eight patients with a chromosomal-encoded ESBL gene (as mentioned above), and one strain with a 54kb ESBL-plasmid (IncN type harboring CTX-M-15) which was not persistent.

In *K. pneumoniae* species complex patients, persistence of the same ESBL-plasmid in different strains within the same patient was less likely, only one patient (out of the three with different strains sharing the same ESBL genes and Inc types) showed whole ESBL-plasmid persistence (Supplementary Fig. S12). On the other hand, in 70% of all eligible *E. coli* patients (with different strains harboring the ESBL gene in plasmids), ESBL-plasmid persistence was evidenced (Supplementary Fig. S13). Six patients were excluded since the ESBL gene was detected in the chromosome.

Out of the seven patients colonized with more than one ESBL-PE species, only two of them followed the criteria to be selected and studied for ESBL-plasmid persistence. In both cases, the persistence of the whole plasmid was confirmed (Supplementary Fig. S14).”

We believe that performing additional long-read sequencing has contributed to the quality and robustness of our findings and thank the reviewer pointing to this shortcoming.

Minor comments:

*** STs for isolates with an unknown ST should be obtained by submitting these to the respective MLST databases. It is a community effort to expand these.**

Response: We agree with the reviewer that it is of particular importance to submit all new sequence types to the corresponding databases. The genomes of *K. pneumoniae* species complex were uploaded to the BIGSdb-Pasteur platform (<https://bigfdb.pasteur.fr/>) and the *E. coli* genomes were submitted to Enterobase (<https://enterobase.warwick.ac.uk/>).

We added the following to the methods section of the manuscript: “*The sequences of isolates of unknown STs were submitted to BIGSdb-Pasteur platform (<https://bigfdb.pasteur.fr/>) and Enterobase (<https://enterobase.warwick.ac.uk/>).*” (page 20, lines 455-456).

*** Hypervirulence in Klebsiella is a great concern. I would suggest that the authors investigate if any of the Klebsiella isolates are hypervirulent. This can be done using the Kleborate tool (<https://github.com/klebgonomics/Kleborate>).**

Response: The authors thank the reviewer for this important suggestion. In response, we performed the virulence analysis with Kleborate. None of the isolates showed a hypervirulent pattern. This was expected since our isolates do not belong to the classical hypervirulent clonal groups already described for *Klebsiella pneumoniae* species complex. A supplementary document was added with the results of this analysis. It reads as follows:

Methods: “Kleborate was used to investigate virulence of the *K. pneumoniae* species complex isolates.” (page 20, lines 440-441).

Results: “Virulence analyses done for all *K. pneumoniae* species complex isolates revealed no hypervirulent strains. A detailed summary of the kleborate results is shown in the Supplementary Material S04.” (page 4, lines 74-76).

*** Line 70-72: Please specify if the “*K. pneumoniae* species” isolates are all *K. pneumoniae* sensu stricto, and which subspecies the “*K. quasipneumoniae*” isolates belong to (similipneumoniae or quasipneumoniae). This can be investigated using Kleborate.**

Response: In response to this reviewer’s comment, we performed additional analyses to determine the subspecies using Kleborate, as suggested by the reviewer. The information was added to the text, tables and supplementary material as follows:

Results: “Among isolates initially classified as *K. pneumoniae* by conventional methods, 70 isolates were confirmed as belonging to *K. pneumoniae* sensu stricto, five were identified as *K. quasipneumoniae* (*K. quasipneumoniae* subsp. *quasipneumoniae* n=3 and *K. quasipneumoniae* subsp. *similipneumoniae* n=2) and one as *K. variicola* by sequencing.” (page 4, lines 71-74).

*** Line 74: The term “*K. pneumoniae* complex” should be “*K. pneumoniae* species complex” to more accurately follow the existing literature. Please change throughout the manuscript and in the Supplementary material.**

Response: We thank the reviewer for pointing this out and changed the term respectively throughout the manuscript and the supplementary material.

*** Line 83: Check Table numbering – Table 2 does not show STs, but AMR genes. Is the ST Table missing from the manuscript?**

Response: We thank the reviewer for identifying this mistake. The table showing information about ST identification is part of the Supplementary Material S02, thus referring to Table 2 was incorrect and this reference was deleted. We only kept the following sentence: “The full list of STs identified is provided in the Supplementary Material S02.”

*** Line 90-91: Is the reference to Fig. 2 correct. I am not able to identify the patients with different clusters in Fig. 2 which basically only show number of days between first and last isolate in each patient. That’s Fig.1**

Response: We apologize for the mistake. The numbering of figures was checked and modified. Indeed, the reference should point to Figure 1, which shows the genetic relatedness of ESBL-PE isolates based on cgMLST, and the different clusters and singletons per patient.

*** Lines 175-176: *oqxA* and *oqxB* is intrinsic genes in *K. pneumoniae* thus the finding of these on the chromosome is not a major finding. Consider to restrict the analysis to acquired resistance genes. It should also be checked if *mdf(A)* is intrinsic genes in *E. coli* and *K. pneumoniae*.**

Response: We thank the reviewer for pointing this out. We restricted the analyses to the acquired resistance genes and removed intrinsic genes from the analysis, as suggested by the reviewer.

Indeed, the *oqxAB* operon, coding for an efflux pump of the RND family, constitute chromosomal intrinsic genes in *K. pneumoniae* (1), existing in most isolates (reviewed in (2)). Expression of this operon in the chromosome is typically residual due to an active near repressor (OqxR) (1). However, it should be noted that *oqxAB* operon can become plasmid borne via transposition events, during which the *oqxAB* genes become overexpressed (presumably as a result of IS26-mediated loss of the OqxR repressor function), which renders a multidrug resistance phenotype (1, 3). Additional transcriptional regulators have been denoted to play a role in expression (2).

It should be noted that, to avoid misinterpretation of the data, we have restricted the search criteria of AMR genes (resfinder) to $\geq 80\%$ coverage and $\geq 80\%$ identity to those in the database. We consider lower thresholds retrieve interspecific hits, as many sequence/base alterations are allowed. With this restricted search parameters, a number of hits vanished, including all *mdfA* hits in *K. pneumoniae*. Yet, *mdfA* is present in all our *E. coli* strains.

Surveillance studies on AMR show that the *mdfA* gene (initially described as *cmr*), an *E. coli* multidrug efflux protein of the MFS family (4, 5), is present in all (6-10) or most (11, 12) *E. coli* isolates; however, *E. coli* expressing MdfA conferring multidrug resistance has been demonstrated in only a few laboratories, always when overexpressed located in plasmids (4, 5, 13-15). Notably, assumptions of drug resistance assigned to *mdfA* when located in the chromosome were not experimentally assessed (7-9, 11, 12). In fact, the *mdfA* gene in the chromosome has been proved to show limited expression level, probably due to lack of complete promoter and RBS region (4, 15) and unchanged drug resistance profile with respect to *mdfA*-deleted variants has been denoted (16, 17). In-depth experimental analyses show a physiological role of MdfA in pH homeostasis (alkaline tolerance) and that multidrug resistance is result of an unrelated adapted second transport function (17). Accumulated scientific evidence suggests that *mdfA* is an old-acquired (codon usage somewhat different from *E. coli* (4)) cryptic gene that *E. coli* adapted, and it confers a multidrug resistance pattern only when it is overexpressed in plasmids (14, 15).

We have changed the methods sections accordingly:

“Intrinsic genes were removed from the analysis when they were exclusively located in the chromosome.” (page 23, lines 507-508). *“Similar to replicon gene prediction, hits with identity and sequence coverage values under 80% were excluded”* (page 21, lines 470-471).

Since *oqxA/B* and *mdfA* were exclusively detected in the chromosome of all PacBio-sequenced isolates, we have decided to remove them from the section of AMR genes located in plasmids (“*Diversity and persistence of acquired resistance genes and Inc replicons*”) (page 9, lines 180). as well as from the respective figure (Supplementary material S01). We indicate now in the methods section that *oqxA/B* and *mdfA* were removed from the analysis as we anticipate they were universally present in all isolates and they were strictly located in the chromosome. We refer to the results section:

Results “*The oqxA/B and mdfA genes were removed from these plasmid analyses as we observed they were intrinsic to the K. pneumoniae and E. coli chromosomes, respectively, as described before in other studies*²⁻⁶.” (page 9, lines 183-185).

To further account for the reviewer’s comment, we looked for further genes potentially located in the chromosome rather than in plasmids. As we performed PacBio sequencing, providing us with a better resolution of the genome, we recognized additional AMR genes located in the chromosome, rather than the plasmids in some of our isolates sequenced by PacBio. To reflect this new finding, within our analyses, we highlighted these genes with an asterisk in the Figures S08-09, and further added a table in Supplementary File 01 which present these genes and their proportion of strains having them in chromosomes rather than in plasmids.

*** Lines 187-188: Table 4 = Table 3? Check labelling of Figures and Tables throughout.**

Response: The numbering of tables and figures was checked and corrected accordingly. We apologize for the mistakes made regarding the labelling.

*** Lines 349-350: Since breakpoints can change the specific version of the EUCAST breakpoint table used should be specified.**

As our study was performed over a time period of ten years, the specific breakpoints applied changed over time. From 01/2008 to 05/2011, breakpoints were interpreted according to the guidelines of the Clinical Laboratory Standards Institute (CLSI), from 06/2011 to 12/2018 breakpoints were interpreted according to the respectively current version of the EUCAST guidelines, ranging from version 1.3 to 8.1.

The Methods section was modified accordingly: “*As our study was performed over a time period of ten years, the specific breakpoints applied changed over time. From 01/2008 to 05/11, minimal inhibitory concentration (MIC) breakpoints were interpreted according to the guidelines of the CLSI, and from 06/11 to 12/2028 breakpoints were interpreted according to the respectively current version of the EUCAST guidelines, ranging from version 1.3 to 8.1 (www.eucast.org).*” (page 17, lines 378-382).

*** Lines 367-368: Specify the number of isolates sequenced using PacBio.**

Response: The required information was added to the text, which now reads as follows:

“To achieve complete genomes consisting of circularized chromosomes and plasmids, 30 isolates were sequenced using the Pacific Biosciences® (PacBio) Sequel I technology (<http://www.pacb.com/>) at the Lausanne Genomic Technologies Facilities (Lausanne, Switzerland).” (page 18, lines 399-401)

“To prove maintenance of the whole ESBL-plasmids over time and between different strains and species within the same patient, additional PacBio sequencing of 95 isolates was performed using the Sequel IIe technology with HiFi reads, at the SeqCenter (Pennsylvania, United States of America).” (page 18, lines 407-410)

*** Line 399-400: Specify the version of PlasmidFinder database used.**

Response: The version of the databases are now specified. The respective section now reads as follows:

“Plasmid replicon genes were predicted by ABRicate v.1.0.1 (<https://github.com/tseemann/abricate>) using the PlasmidFinder database (accessed on July 1st 2021)” (page 21, lines 458-459).

*** Line 407-408: Specify the version of ResFinder database used.**

Response: The version of the databases are now specified. The respective section now reads as follows

“For this, AMR genes were predicted by ABRicate v.1.0.1 (<https://github.com/tseemann/abricate>) using the Resfinder database (accessed on July 1st 2021).” (page 21, lines 468-470).

*** Supplementary material: There is two Tables labelled with “Table S1”**

Response: The numbering of tables was checked and modified, accordingly.

We thank the reviewers for their valuable comments, which we feel have lead to a much improved version of the manuscript.

Sincerely,

Dr. Sarah Tschudin-Sutter (on behalf of all co-authors)

References

1. Tacconelli E, Mazzaferri F, de Smet AM, et al. ESCMID-EUCIC clinical guidelines on decolonization of multidrug-resistant Gram-negative bacteria carriers. *Clin Microbiol Infect* 2019; **25**(7): 807-17.
1. Tacconelli E, Mazzaferri F, de Smet AM, et al. ESCMID-EUCIC clinical guidelines on decolonization of multidrug-resistant Gram-negative bacteria carriers. *Clin Microbiol Infect* 2019; **25**(7): 807-17.
2. Wong MH, Chan EW, Chen S. Evolution and dissemination of OqxAB-like efflux pumps, an emerging quinolone resistance determinant among members of Enterobacteriaceae. *Antimicrobial agents and chemotherapy* 2015; **59**(6): 3290-7.
3. Li J, Zhang H, Ning J, et al. The nature and epidemiology of OqxAB, a multidrug efflux pump. *Antimicrob Resist Infect Control* 2019; **8**: 44.
4. Yasir M, Farman M, Shah MW, et al. Genomic and antimicrobial resistance genes diversity in multidrug-resistant CTX-M-positive isolates of Escherichia coli at a health care facility in Jeddah. *J Infect Public Health* 2020; **13**(1): 94-100.

5. Morita S, Sato S, Maruyama S, et al. Whole-genome sequence analysis of Shiga toxin-producing *Escherichia coli* O157 strains isolated from wild deer and boar in Japan. *J Vet Med Sci* 2021; **83**(12): 1860-8.
6. Sahin S, Mogulkoc MN, Kurekci C. Disinfectant and heavy metal resistance profiles in extended spectrum beta-lactamase (ESBL) producing *Escherichia coli* isolates from chicken meat samples. *Int J Food Microbiol* 2022; **377**: 109831.

REVIEWERS' COMMENTS

Reviewer #2 (Remarks to the Author):

The authors have adequately addressed my comments and significantly improved the manuscript. I appreciate the effort that the authors have done with expanding significantly the number of isolates subjected to long-read sequencing and integrated these results in the manuscript.